# Divergent changes in particulate and mineral-associated organic carbon upon permafrost thaw

Futing Liu[1,2], Shuqi Qin[2,3], Kai Fang[2,3], Leiyi Chen [2], Yunfeng Peng [2], Pete Smith [4] & Yuanhe Yang [2,3] ✉

Permafrost thaw can stimulate microbial decomposition and induce soil carbon (C) loss, potentially triggering a positive C-climate feedback. However, earlier observations have concentrated on bulk soil C dynamics upon permafrost thaw, with limited evidence involving soil C fractions. Here, we explore how the functionally distinct fractions, including particulate and mineral-associated organic C (POC and MAOC) as well as iron-bound organic C (OC-Fe), respond to permafrost thaw using systematic measurements derived from one permafrost thaw sequence and five additional thermokarst-impacted sites on the Tibetan Plateau. We find that topsoil POC content substantially decreases, while MAOC content remains stable and OC-Fe accumulates due to the enriched Fe oxides after permafrost thaw. Moreover, the proportion of MAOC and OC-Fe increases along the thaw sequence and at most of the thermokarst-impacted sites. The relatively enriched stable soil C fractions would alleviate microbial decomposition and weaken its feedback to climate warming over long-term thermokarst development.

Permafrost occupies ~16% of the global terrestrial area[1] but contributes ~50% of the soil organic carbon (C) stock in terrestrial ecosystems[2], with the permafrost zone in the Northern Hemisphere storing as much as 1014 Pg (1 Pg = $10^{15}$ g) C in the top 3 m[3]. This soil C stock is roughly twice that of living vegetation and 1.5 times higher than the atmospheric C pool[3,4]. Even slight changes in this large permafrost C stock could exert significant impacts on the concentration of atmospheric carbon dioxide ($CO_2$)[5]. In particular, warming-induced permafrost thaw would cause substantial soil C emissions, which could further induce a positive permafrost C feedback to climate warming[5–7]. It has been reported that thawing permafrost would result in 1.0–1.7 Pg C release per year over the next 300 years[8,9], triggering an additional increase of global air temperature by 0.20–0.44 °C[8,10]. A comprehensive evaluation of how thawing permafrost alters the soil C pool is, therefore, of the utmost

importance for predicting the direction and magnitude of permafrost C-climate feedback[11].

Given the critical role of the permafrost C pool in affecting terrestrial C-climate feedbacks, soil C variations in permafrost regions have received considerable attention from the global change research community[5,6,8,11]. Site-level observations have reported that permafrost thaw could lead to a large amount of soil C loss within years and decades[12–14], but substantial uncertainties exist in the model-projected direction and magnitude of soil organic C (SOC) changes under various climate change scenarios[8,15]. Much of the uncertainty might be ascribed to the partitioning of soil C into conceptual model pools that are not biophysically defined, without regard to the shift in soil C fractions with time[10,16]. Improved understanding of the dynamics of soil C fractions will enable a more precise prediction of soil C vulnerability to external disturbances[17,18] such as permafrost thaw[19]. Generally, bulk soil

[1]Key Laboratory of Forest Ecology and Environment of National Forestry and Grassland Administration, Ecology and Nature Conservation Institute, Chinese Academy of Forestry, 100091 Beijing, China. [2]State Key Laboratory of Vegetation and Environmental Change, Institute of Botany, Chinese Academy of Sciences, 100093 Beijing, China. [3]University of Chinese Academy of Sciences, 100049 Beijing, China. [4]Institute of Biological and Environmental Sciences, University of Aberdeen, Aberdeen AB24 3UU, UK. ✉e-mail: yhyang@ibcas.ac.cn

C could be divided into particulate organic C (POC) and mineral-associated organic C (MAOC) with different formation, function, and turnover times[16–18]. Of them, POC is a mixture of partially decomposed plants and their decomposition byproducts, which is vulnerable to microbial decomposition and has a short mean residence time (<10 years·decades)[18,20]. By comparison, MAOC is formed by combining small molecules of organic C leached from plants or transformed by microorganisms with soil minerals[16–18,21,22]. This C fraction is protected by the chemical bonds associated with minerals and belongs to the stable C component with a long-term turnover time (decades-centuries)[18]. Particularly, iron-bound organic C (OC-Fe), a critical component of MAOC, favors the long-term preservation of SOC[23] since iron oxides can adsorb certain functional groups of SOC[24,25] and constrain the enzyme activity of soil microorganisms[26,27]. Consequently, soil C emissions could be reduced if the amount of MAOC or OC-Fe increases under permafrost thaw. Accordingly, changes in these mineral-associated organic C fractions would strengthen the stability of the entire soil C pool and weaken the potential permafrost C-climate feedback[16]. Nevertheless, limited evidence is available on the responses of POC and MAOC as well as OC-Fe components to permafrost thaw since previous studies primarily focused on bulk soil C[12,14].

In this study, we seek to test the following three hypotheses: (1) POC content may exhibit a substantial decline due to the improved soil aeration and microbial C processing caused by abrupt permafrost thaw occurring in uplands like thermo-erosion gullies[5,12]. (2) MAOC content, which is difficult for decomposers to degrade due to the adsorption of organic C to soil minerals[18,28], would remain stable upon permafrost thaw. (3) OC-Fe content might increase due to the improvement of soil aeration and subsequent increase in the oxidation of Fe(II) to Fe(III)[23]. To test these hypotheses, we first analyzed changes of topsoil POC, MAOC, and OC-Fe along a permafrost thaw sequence (i.e., non-collapse and 1, 10, and 16 years since collapse induced by a thermo-erosion gully; collapse time is relative to the sampling year of 2014) on the Tibetan Plateau[13]. To verify the generality of observations along this permafrost thaw sequence, we then examined changes in topsoil C fractions in response to permafrost collapse by selecting another five sites (also characterized as thermo-erosion gullies) along a 550 km permafrost transect (Fig. 1). With the combination of observations from one permafrost thaw sequence together with five thermokarst-impacted sites over the regional scale, we demonstrate that OC-Fe becomes enriched upon permafrost collapse despite the relatively stable MAOC and the substantially reduced POC fractions, accompanied with an increased proportion of MAOC and OC-Fe at most of the thermokarst-impacted sites.

## Results

### Permafrost thaw-induced changes in soil C fractions

SOC content in the top 15 cm gradually decreased with the collapse time, which was significantly lower than that in the control after 10 years of permafrost collapse ($P < 0.05$; Fig. 2a). Compared with the non-collapsed control, the thermo-erosion gully induced SOC loss by 29.4% over 16 years. Similar to bulk soil C, the contents of POC and heavy POC gradually decreased (all $P < 0.05$; Fig. 2b, c), while that of MAOC showed no significant changes along the thaw sequence

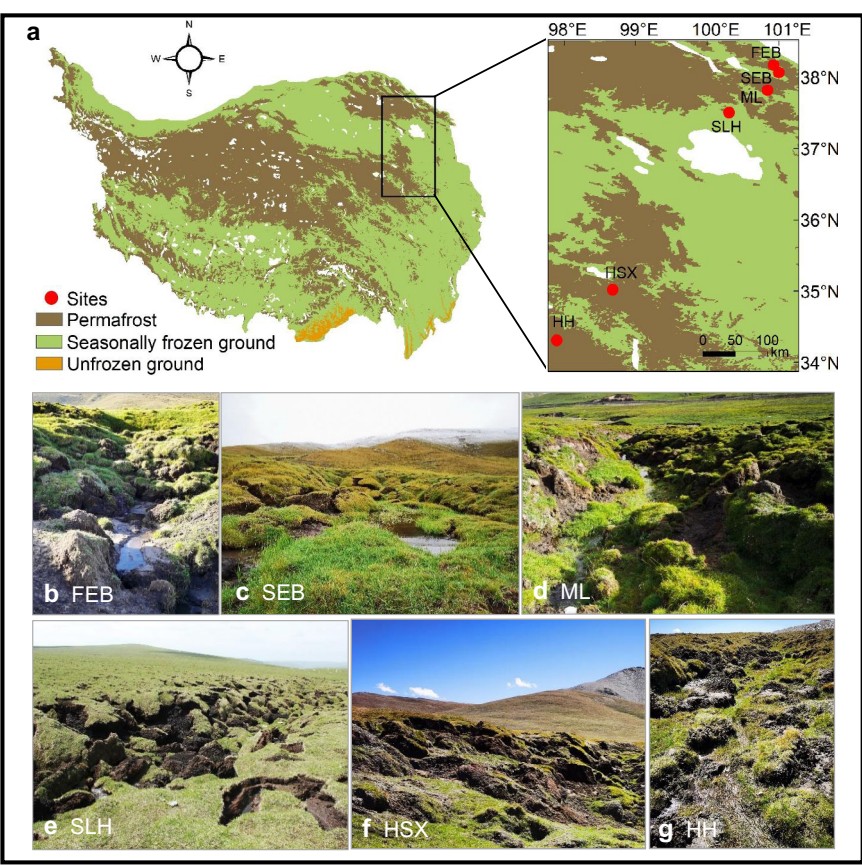

**Fig. 1 | Sampling sites and landscapes of upland thermokarst on the Tibetan Plateau. a–g** Sampling location of the six sites shown on the map of permafrost distribution on the Tibetan Plateau[64] (**a**) and individual features of thermokarst-impacted sites such as FEB (**b**), SEB (**c**), ML (**d**), SLH (**e**), HSX (**f**), and HH (**g**) (photo credit for panels **b**–**g**: F.T. Liu). Notably, the sites involved in this study were mainly distributed across the eastern plateau since upland thermokarst primarily occurred in this area. FEB, the first site at Ebo; SEB, the second site located at Ebo; ML, SLH, HSX, and HH indicate the sites at Mole, Shaliuhe, Huashixia, and Huanghe, respectively. The map images (panel **a**) were created by authors using ArcMap 10.2 (Environmental Systems Research Institute, Inc., Redlands, CA, USA) /[64] (https://tc.copernicus.org/articles/11/2527/2017/) / CC BY (https://creativecommons.org/licenses/by/3.0/).

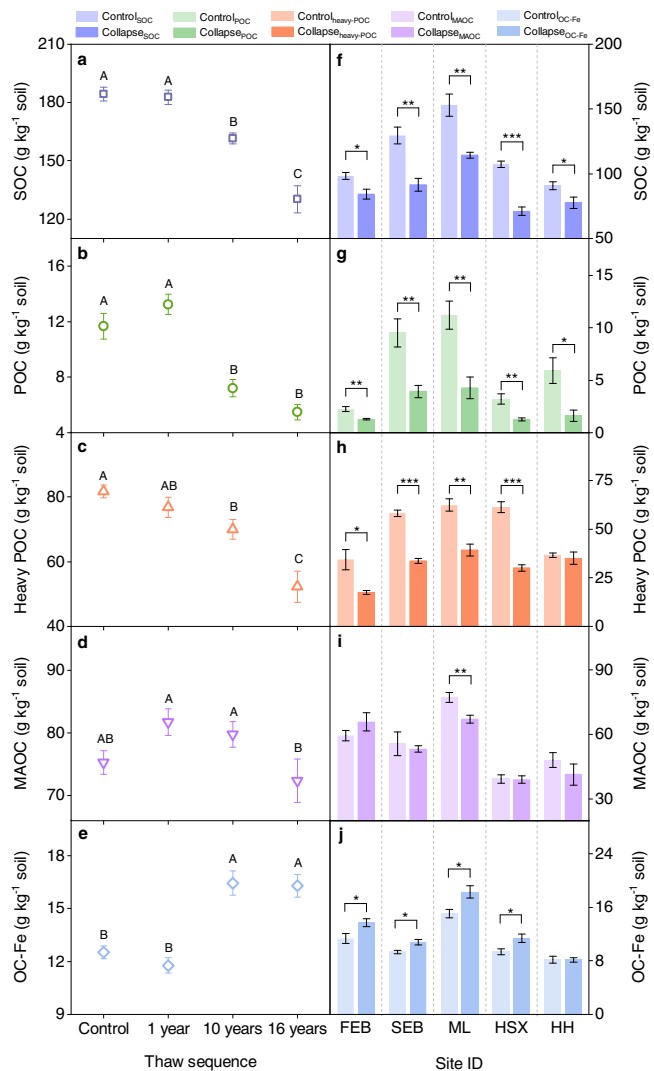

**Fig. 2 | Changes of SOC and its fractions induced by permafrost collapse.**
**a–e** Changes in contents of SOC (**a**), POC (**b**), heavy POC (**c**), MAOC (**d**), and OC-Fe (**e**) along a permafrost thaw sequence. **f–j** Comparisons of SOC (**f**), POC (**g**), heavy POC (**h**), MOAC (**i**), and OC-Fe (**j**) between collapsed (dark colour) and control (light color) plots at each of the five regional thermokarst-impacted sites: FEB, SEB, ML, HSX, and HH. SOC, soil organic carbon; POC, particulate organic carbon (density < 1.6 g cm⁻³); Heavy POC, heavy particulate organic carbon (density > 1.6 g cm⁻³ and size > 53 μm); MAOC, mineral-associated organic carbon (density > 1.6 g cm⁻³ and size < 53 μm); OC-Fe, iron-bound organic carbon; FEB, the first site at Ebo; SEB, the second site located at Ebo; ML, HSX, and HH indicate the sites at Mole, Huashixia, and Huanghe, respectively. Error bars represent standard errors. Different capital letters indicate significant differences for the variables within plots along the thaw sequence (LSD test, $P < 0.05$). Dashed lines distinguish different thermokarst-impacted sites, denoting that the parameters are compared between collapsed and control plots in each site rather than across various study sites. SOC data along the thaw sequence were reanalyzed from ref. 13. *$P < 0.05$, **$P < 0.01$, and ***$P < 0.001$.

($P > 0.05$; Fig. 2d). These observations were confirmed by data from five additional sites at the regional scale. The occurrence of thermo-erosion gullies significantly decreased SOC content at the five thermokarst-impacted sites (all $P < 0.05$; Fig. 2f), with C losses of 14.3%, 29.5%, 25.3%, 33.7%, and 14.3%, respectively. Furthermore, both POC and heavy POC contents declined at these sites after permafrost collapse (all $P < 0.05$; Fig. 2g, h) except for one site at Huanghe, showing no change in the heavy POC. By comparison, the MAOC content did not exhibit significant changes before and after permafrost collapse at four of the five sampling sites (all $P > 0.05$; Fig. 2i). Despite the relatively stable MAOC, the OC-Fe content significantly increased after

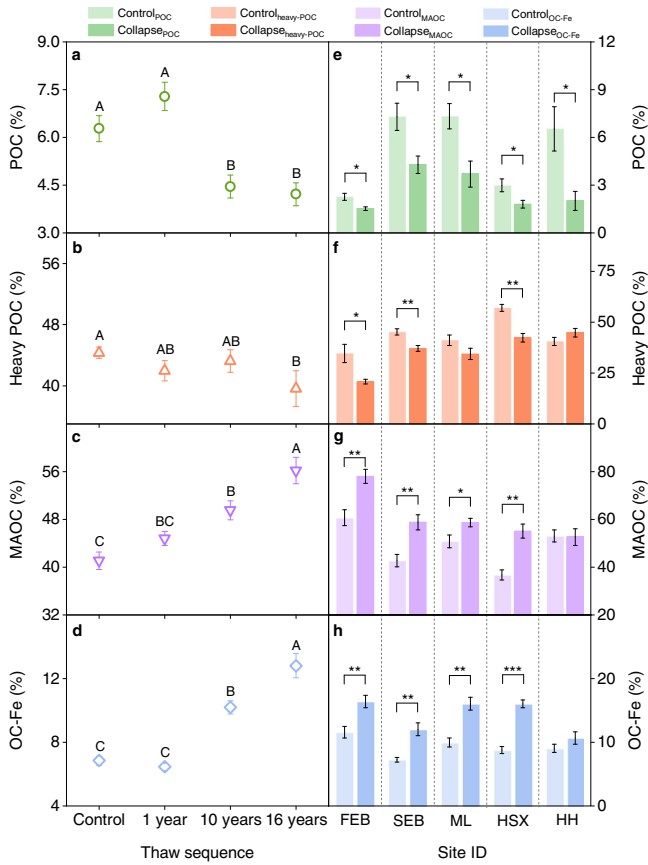

**Fig. 3 | Changes in the proportion of soil C fractions to SOC after permafrost collapse. a–d** Variations of the proportion of POC (**a**), heavy POC (**b**), MAOC (**c**), and OC-Fe (**d**) to SOC along the thaw sequence. **e–h** Comparisons of the percentage of POC (**e**), heavy POC (**f**), MAOC (**g**), and OC-Fe (**h**) between collapsed (dark color) and control (light colour) plots in each site at the regional scale. SOC, soil organic carbon; POC, particulate organic carbon (density < 1.6 g cm⁻³); Heavy POC, heavy particulate organic carbon (density > 1.6 g cm⁻³ and size > 53 μm); MAOC, mineral-associated organic carbon (density > 1.6 g cm⁻³ and size < 53 μm); OC-Fe, iron-bound organic carbon; FEB, the first site at Ebo; SEB, the second site located at Ebo; ML, HSX, and HH indicate the sites at Mole, Huashixia, and Huanghe, respectively. Error bars represent standard errors. Different capital letters indicate significant differences for the variables within plots along the thaw sequence (LSD test, $P < 0.05$). Dashed lines distinguish different thermokarst-impacted sites, denoting that the parameters are compared between collapsed and control plots in each site rather than across various study sites. *$P < 0.05$, **$P < 0.01$, and ***$P < 0.001$.

permafrost collapse, both along the thaw sequence and at four of the five regional-scale sites (all $P < 0.05$; Fig. 2e, j). These results illustrated that SOC loss was largely driven by the decrease of POC and heavy POC, while MAOC remained stable and OC-Fe continuously accumulated.

The proportion of POC to SOC significantly decreased after permafrost collapse at all of the thermokarst-impacted sites (all $P < 0.05$; Fig. 3a, e). Similarly, the proportion of heavy POC declined along the thaw sequence and at the three sites from Ebo and Huashixia (all $P < 0.05$; Fig. 3b, f). By comparison, the proportion of MAOC and OC-Fe to SOC significantly increased along the thaw sequence and at four of the five sites at the regional scale (all $P < 0.05$; Fig. 3c, d, g, h) except for one site at Huanghe. Changes in the proportion of soil C fractions demonstrated that MAOC became a major fraction of SOC at the late stage of permafrost collapse (Supplementary Table 1) with a significant increment of OC-Fe, strengthening the stability of SOC. Additionally, the carbon: nitrogen ratio of particulate organic matter significantly decreased along the thaw sequence and at four of the five sites over the regional scale (all $P < 0.05$; Supplementary Table 1). By comparison, the

carbon: nitrogen ratio in mineral-associated organic matter significantly increased along the thaw sequence and at the two sites at Ebo (all $P < 0.05$), while the ratio at the other three sites showed no changes after permafrost collapse (all $P > 0.05$; Supplementary Table 1).

## Changes in biotic and abiotic variables upon permafrost thaw

The formation of thermo-erosion gullies altered vegetation biomass (Supplementary Fig. 1a, b). Specifically, along the thaw sequence located at Shaliuhe, both aboveground and belowground biomass significantly increased in the 10 years since permafrost collapse (all $P < 0.05$; Supplementary Fig. 1a, b), although there were no significant changes in the early or late stages of permafrost collapse (collapsed for 1 year and 16 years, respectively; all $P > 0.05$). Four of the five additional sites at the regional scale, those at Ebo, Mole, and Huashixia, showed no obvious changes in either aboveground or belowground biomass after permafrost collapse (all $P > 0.05$; Supplementary Fig. 1a, b). However, at the Huanghe site, belowground biomass significantly increased ($P < 0.05$; Supplementary Fig. 1b) while aboveground biomass did not show any significant variations ($P > 0.05$; Supplementary Fig. 1a).

Permafrost collapses also led to substantial changes in topsoil physical and chemical properties as well as soil minerals (Supplementary Fig. 1c–l). For the case of soil properties, permafrost collapse significantly elevated soil temperature and pH (all $P < 0.05$; Supplementary Fig. 1c, e), while decreasing soil moisture (all $P < 0.05$; Supplementary Fig. 1d). The percentage of clay and silt significantly decreased along the thaw sequence and at the four sites in Ebo, Mole, and Huashixia (all $P < 0.05$), whereas no significant change was

observed at the Huanghe site ($P > 0.05$; Supplementary Fig. 1f). With respect to soil minerals, the content of Fe oxides, including pedogenic Fe oxides, poorly crystalline Fe oxides and organically complexed Fe oxides, significantly increased along the thaw sequence and at all of the thermokarst-impacted sites (all $P < 0.05$; Supplementary Fig. 1g–i) except for Huanghe where there was no change in the complexed Fe oxides ($P > 0.05$; Supplementary Fig. 1i). Clay minerals mainly consisted of illite, mixed-layer illite/smectite, chlorite and kaolinite in our study sites (Supplementary Table 2), and the former two accounted for more than 80% of clay mineral composition (Supplementary Fig. 1j, k). However, both the illite and mixed-layer illite/smectite minerals showed no significant changes along the thaw sequence or at the five thermokarst-impacted sites over the regional scale (all $P > 0.05$; Supplementary Fig. 1j, k). Furthermore, the specific surface area of soil mineral-associated organic matter showed no significant changes along the thaw sequence or at the four sites at Ebo, Huashixia, and Huanghe (all $P > 0.05$; Supplementary Fig. 1l), although there was a significant decrease at the Mole site ($P < 0.05$).

## Linkages of soil C fractions with biotic and abiotic factors upon permafrost thaw

Along the thaw sequence, POC content was significantly correlated with both biotic (aboveground biomass) and abiotic (soil temperature, moisture, and pH) variables (all $P < 0.05$; Supplementary Fig. 2). To explore the relative importance of these biotic and abiotic factors in influencing POC content, we performed a random forest modeling analysis to screen the above factors and observed that soil moisture was the most important factor influencing POC (Fig. 4a). Specifically,

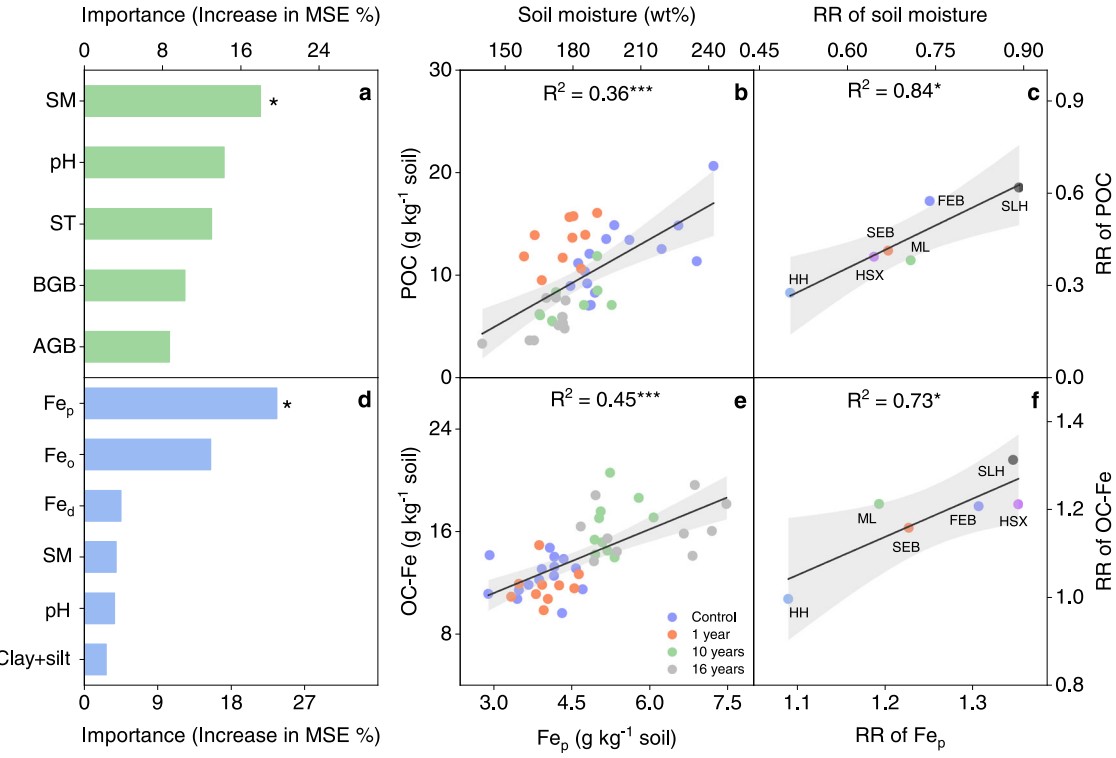

**Fig. 4 | Relationships of POC and OC-Fe contents with potential drivers.**
**a, b** Relative importance of biotic and abiotic variables in influencing POC revealed by random forest model (**a**) and relationship of POC content with soil moisture (**b**) along the thaw sequence. **c** Relationship of response ratio (RR, the ratio of two variables between collapsed and control plots [Variable$_{collapse}$/Variable$_{control}$]) of POC with RR of soil moisture across the SLH, FEB, SEB, ML, HSX, and HH sites on the Tibetan Plateau. **d, e** Relative importance of edaphic variables and various Fe oxides in affecting OC-Fe (**d**) and association between OC-Fe and Fe$_p$ (**e**) along the thaw sequence. **f** Linkages of RR of OC-Fe with RR of Fe$_p$. POC, particulate organic

carbon; OC-Fe, iron-bound organic carbon; SM, soil moisture; ST, soil temperature; BGB, belowground biomass; AGB, aboveground biomass; Fe$_p$, organically complexed Fe oxides; Fe$_o$, poorly crystalline Fe oxides; Fe$_d$, pedogenic Fe oxides; Clay + silt, the percentage of clay and silt; MSE, mean squared error; FEB, the first site at Ebo; SEB, the second site located at Ebo; ML, SLH, HSX, and HH indicate the sites at Mole, Shaliuhe, Huashixia, and Huanghe, respectively. The linear regression lines with 95% confidence intervals represent the predicted effects of fixed factors. *$P < 0.05$, and ***$P < 0.001$.

POC content was positively associated with soil moisture ($P < 0.001$; Fig. 4b), reflecting the fact that soil drainage could lead to a decrease in the POC component along the thaw sequence. Different from POC fraction, the most likely driver of OC-Fe was organically complexed Fe oxides (Fig. 4d), despite significant correlations of OC-Fe with soil moisture, pH, pedogenic Fe oxides, and poorly crystalline Fe oxides (all $P < 0.05$; Supplementary Fig. 2). The organically complexed Fe oxides were positively correlated with OC-Fe ($P < 0.001$; Fig. 4e), indicating that the increased Fe oxides and improved soil aeration induced by permafrost collapse could result in the accumulation of OC-Fe along the thaw sequence.

To verify the universality of our observations along the thaw sequence, we analyzed the effects of the above-mentioned factors on the variations of POC and OC-Fe induced by permafrost thaw across the regional sites. By examining the correlations between the response ratio (RR, the ratio of two variables between collapsed and control plots [Variable$_{collapse}$/Variable$_{control}$]) of POC (or OC-Fe) and explanatory variables (Fig. 4c, f), we obtained similar patterns across the regional sites to those observed along the thaw sequence. Permafrost thaw-induced shifts in POC content were positively correlated with the corresponding changes in soil moisture ($P < 0.05$; Fig. 4c), and permafrost thaw-induced dynamics in OC-Fe content were closely associated with the corresponding variations in organically complexed Fe oxides ($P < 0.05$; Fig. 4f). Overall, these observations derived from both the thaw sequence and the regional thermokarst-impacted sites consistently demonstrated close linkages between the POC component and soil moisture and between the OC-Fe content and Fe oxides.

## Discussion

Our results revealed that permafrost collapse resulted in significant C loss in the surface soil (Fig. 2a, f). Considering that aboveground and belowground biomass either did not significantly change or showed a slight increase after permafrost collapse (Supplementary Fig. 1a, b), exogenous plant C inputs seemed to exert limited effects on SOC loss. Due to this point, the decline of SOC content may be largely due to the accelerated C output *via* microbial decomposition and lateral transport. On the one hand, permafrost collapse occurring in upland regions usually alters the in situ anaerobic state and improves soil aeration due to soil drainage[12,13]. The improved soil aeration can then alleviate the oxygen limitation to microorganisms[29,30], and thus elevate the availability of electron acceptors for microbial decomposition[31], ultimately accelerating soil organic matter decomposition[12,32–34]. On the other hand, the occurrence of thermo-erosion gullies is generally accompanied by water tracks or headwater streams[35], which is also the case in our study sites, both along the thaw sequence and at the regional thermokarst-impacted sites. The formation of a headwater stream along a thermo-erosion gully would inevitably result in soil C loss through lateral C transport to the surrounding water[36–38]. Taken together, microbial degradation and lateral transport may be the two major pathways responsible for the substantial soil C loss observed upon permafrost collapse.

Despite soil C loss, POC and MAOC components exhibited divergent responses to permafrost collapse, with a significant decrease in the former while relatively stable for the latter, supporting our initial hypotheses. These distinct responses of the POC and MAOC components could be attributed to fundamental differences in the degree of soil C protection between these two fractions. Specifically, POC is largely composed of lightweight fragments that are not protected by mineral associations, which is thus more vulnerable to microbial decomposition[18,20,39,40]. Consistent with this deduction, positive associations were observed between POC content and soil moisture along the thaw sequence and also between thaw-induced shifts in POC content and the corresponding changes in soil moisture across the regional thermokarst-impacted sites (Fig. 4b, c), reflecting that the decreased soil moisture and improved soil aeration induced by permafrost collapse could stimulate microbial decomposition and thus lead to the POC loss. By comparison, MAOC is protected by Fe oxides and clay minerals through sorptive interactions[16,18,39,41]. In particular, clay-sized particles, including Fe oxides and phyllosilicates, can provide a high specific surface area and thus adsorb substantial amounts of SOC[23,42,43]. These organo-mineral complexes could stabilize soil C through the formation of chemical bonds and subsequently shield SOC against microbial decomposition[23,42–45]. Consequently, the different degrees of mineral protection could lead to the situation that MAOC could persist for much longer than POC[18,46–48].

Apart from their distinct protection statuses, the density differences between POC and MAOC could also lead to a larger loss of POC through lateral transport. Given that POC has a lower density than MAOC[18], it is more vulnerable to leaching than MAOC because of the preferential transport of light organic particles[49,50]. Nevertheless, although MAOC is more stable than POC, over long-term scales (i.e., decadal or centennial timescales), it would also be decomposed and laterally transferred[50,51]. However, this part of the MAOC loss might be offset by increased microbial residual C (an important source of MAOC)[18,39,52] either along the thaw sequence or at most of the regional thermokarst-impacted sites (Supplementary Fig. 3), thus resulting in a relatively stable MAOC content. This possibility needs to be tested by conducting an isotope labeling experiment (add $^{13}$C labeled plant material to soil) to trace the formation and loss of MAOC[22,40,53,54] in the near future. Collectively, the functional differences between POC and MAOC may lead to their divergent responses to permafrost collapse.

In support of our third hypothesis, both the OC-Fe content and its proportion to SOC significantly increased after permafrost collapse, both along the permafrost thaw sequence and at most of the thermokarst-impacted sites over the regional scale. Specifically, the amount of OC-Fe and its proportion to SOC increased significantly by 21.9% and 67.5% after permafrost collapse, respectively (Figs. 2 and 3). Our findings of enriched OC-Fe after permafrost collapse were supported by earlier observations across the study area[55,56], which jointly illustrated widespread accumulation in OC-Fe after upland thermokarst formation. Interestingly, this enrichment in OC-Fe after upland thermokarst distinctly differs from the observations from lowland thermokarst in the Arctic, which have revealed reduced OC-Fe upon permafrost thaw[28]. Such a discrepancy could be largely due to the different trajectories of soil moisture after thermokarst formation[56–59], which acts as a critical driver for Fe(III) reduction and Fe(II) oxidation[28,58–60]. Specifically, water-logging is generally induced in lowland regions[28,61,62], while soil drainage often occurs in upland areas upon permafrost collapse[12,13,63]. It has been demonstrated that water-logging favors the reductive dissolution of reactive Fe(III) minerals (also referred to as Fe oxides) because Fe(III)-bearing minerals can be used as electron acceptors for anaerobic respiration by Fe(III)-reducing microorganisms[23,28,59]. Given that reactive Fe(III) minerals could stabilize SOC via sorption/co-precipitation and protect it from microbial degradation[23,24,43], the Fe(III) reduction occurring in lowland thermokarst could result in the loss of OC-Fe. Conversely, in upland thermokarst, with the improvement of aeration after thermokarst formation, oxygen can replace Fe(III) as electron acceptors for microbial respiration and also stimulate the Fe(II) oxidation to Fe(III), thereby promoting the formation of OC-Fe associations[23,59]. In support of this deduction, Fe oxides were observed to increase significantly after thermokast formation at most of our study sites (Supplementary Fig. 1g–i). Moreover, both Fe oxides and OC-Fe were negatively correlated with soil moisture (Supplementary Fig. 4), which jointly demonstrated that soil drainage after upland thermokarst formation was the main reason for the accumulation of Fe oxides and OC-Fe. Consequently, the improvement of soil aeration induced by permafrost collapse in upland regions could elevate the content of Fe oxides and enhance the protective effects of Fe phases on soil C pools, ultimately resulting in the accumulation of OC-Fe.

In summary, with the combination of observations from a thaw sequence and five additional sites over the regional scale, we observed that the formation of thermo-erosion gullies led to divergent dynamics of various topsoil C fractions. Among them, the amount of POC decreased substantially while the MAOC content remained stable and OC-Fe accumulated after permafrost collapse, resulting in a significant increase in the proportions of MAOC and OC-Fe along the thaw sequence and at most of the regional thermokarst-impacted sites. The relatively enriched stable soil C fractions illustrate the enhanced soil C stability, which may alleviate soil C emissions and weaken the permafrost C-climate feedback during long-term thermokasrt development. Meanwhile, the accumulation of OC-Fe with the improvement of soil aeration after the formation of thermo-erosion gullies demonstrates the pivotal role of Fe oxides in affecting soil C stabilization in upland permafrost regions. Overall, these findings highlight the importance of incorporating this critical mineral-associated organic C fraction into Earth system models to accurately predict the permafrost C cycle in response to climate warming.

## Methods

### Study sites, experimental design, and field sampling

The Tibetan alpine permafrost region, the largest area of permafrost in the middle and low latitudes of the Northern Hemisphere[64], stores substantial soil C (15.3–46.2 Pg C within 3 m depth)[65–67]. With continuous climate warming, permafrost thaw has triggered the formation of widespread thermokarst landscapes across this permafrost area[13,68]. To explore the impacts of thermokarst formation and development on soil C dynamics, we collected topsoil samples (0–15 cm) from a thaw sequence in 2014 and from five additional sites spread across the region in 2020. The thermokarst landscape was characterized as thermo-erosion gullies (Supplementary Table 2). The elevation of these six sites is between 3515 and 4707 m. The mean annual temperature across this area ranges from −3.1 to 2.6 °C, and the average annual precipitation varies from 353 to 436 mm. The vegetation type across these sites is swamp meadow, with the dominant species being *Kobresia tibetica, Kobresia royleana* and *Carex atrofuscoides*. Although the dominant species did not change after permafrost collapse, the forb coverage increased along the thaw sequence and across the five additional thermokarst-impacted sites. The main soil type is Cryosols on the basis of the World Reference Base for Soil Resources[69], with soil pH ranging from 5.6 to 7.3 (Supplementary Fig. 1e). The active layer thickness varies between 0.7 and 1.1 m across the six study sites and the underlying soil parent material is either siliciclastic sedimentary or unconsolidated sediments (Supplementary Table 2).

To evaluate the dynamics of soil C fractions after permafrost collapse, we collected soil samples across the Tibetan alpine permafrost region based on the following two steps (Supplementary Fig. 5). In the first step, we established six collapsed plots (-15 × 10 m) along a thaw sequence (located in Shaliuhe close to Qinghai Lake, Qinghai Province, China), which had been collapsed for 1, 3, 7, 10, 13, and 16 years before the sampling year of 2014[13]. The collapse time of each plot was estimated by dividing the distance between the collapsed plot and the gully head by the retreat rate (-8.0 m year⁻¹; the rate of the head-wall retreat was determined by Google Earth satellite images and in situ monitoring)[13]. Then, we set up six paired control (non-collapsed) plots adjacent to these collapsed plots. To limit experimental costs, we selected three paired control and collapsed plots (collapsed for 1, 10, and 16 years, representing the early, middle, and late stages of collapse) to examine the responses of POC, MAOC and OC-Fe to permafrost collapse (Supplementary Fig. 5). Within each collapsed plot, we collected topsoil (0–15 cm) samples from all vegetated patches (Supplementary Fig. 6), and then evenly selected 10 vegetated patches for this study considering the heavy workload and high cost. In each selected vegetated patch, 5–8 soil cores were sampled and completely mixed as one replicate. Within each control plot, topsoil samples were

randomly collected from five quadrats at the center and four corners of the plot. In each quadrat, 15–20 soil cores were sampled and mixed as one replicate. Thereby, ten replicates were acquired in each collapsed plot ($n = 10$), and five replicates were obtained in each control plot ($n = 5$). In total, we acquired 45 soil samples, including 30 samples from the three collapsed plots and 15 samples from the non-collapsed control for subsequent analysis.

In the second step, to further verify the universality of collapse effects on SOC fractions, we collected topsoil (0–15 cm) samples from an additional five similar sites located near the towns of Ebo, Mole, Huashixia, and Huanghe across a 550 km permafrost transect in August 2020 (Fig. 1). Specifically, paired collapsed and control plots (15 × 10 m) were established at the end of a gully and in adjacent non-collapsed areas in each site (Supplementary Fig. 5). In the collapsed plot, we set five 5 × 3 m quadrats at the center and four corners of the plot, and then collected topsoil samples within all the vegetated patches in these quadrats. In each quadrat, all the collected soil cores (15–20 cores) were completely mixed as one replicate, and finally, five replicates were acquired in each collapsed plot ($n = 5$). Similarly, five replicates were obtained from the five quadrats in each control plot ($n = 5$). In total, we collected 50 topsoil samples across these five thermokarst-impacted sites. After transportation to the laboratory, all the soil samples were handpicked to remove surface vegetation, roots and gravels, and sieved (2 mm) for subsequent analysis.

It should be noted that the space for time approach was only used for the permafrost thaw sequence, not for the other five sites over the regional scale. Across these five sites, we focused on the impact of permafrost collapse on POC, MAOC as well as OC-Fe by comparing soil C fractions inside and outside the gully in each site rather than among the study sites. Given the low coefficient of variation of parameters (i.e., edaphic variables and soil minerals) in the control plot of each site (Supplementary Table 3), the pristine soils in each site could also be regarded as homogeneous[70], and the differences in parameters inside and outside the gully could be attributed to the effects of permafrost collapse. Along the permafrost thaw sequence, to verify whether the plots with different collapse times (1, 10, and 16 years) were comparable, we analyzed a series of parameters (i.e., vegetation biomass, edaphic variables, and soil minerals) for the three control plots which were located outside the gully but adjacent to three collapsed plots within the gully (Supplementary Fig. 5). By comparing aboveground biomass, belowground biomass, SOC, soil moisture, pH, bulk density, soil texture, and soil minerals (see below for details of the analytical method), we observed that the above parameters were not significantly different among the three control plots along the thaw sequence (all $P > 0.05$; Supplementary Fig. 7). These comparisons demonstrated that the study area was homogeneous before permafrost thaw and thus it was reasonable to adopt the space for time approach along the permafrost thaw sequence.

It should also be noted that the collected topsoil samples used in this study were less affected by physical mixing and translocation due to thaw phenomena at the thermokarst-impacted sites. Specifically, to examine changes in soil properties upon permafrost thaw, we chose to collect topsoil within the vegetated patches rather than from the exposed soil areas in the collapsed plots (Supplementary Fig. 6). These vegetated patches (40–60 cm thickness) are formed during the landscape fragmentation after permafrost collapse[13]. Although permafrost collapse inevitably led to soil translocation, these vegetated patches maintained their original shapes, especially for the topsoil because it is protected by mattic epipedon in this swamp meadow ecosystem on the Tibetan Plateau (which has an intensive root network protecting soils against interference)[71,72]. Moreover, we collected 0–15 cm of topsoil within the vegetated patches, in which soil cores were at least 10 cm away from the edge of the patch. Due to these two points, topsoil should not be mixed with the subsoil in our case. To test this deduction, we compared the non-collapsed (control) plot with the

collapsed plot occurring for 1 year (the early stage of the permafrost thaw sequence), and observed no significant differences in soil properties such as bulk density, SOC, pH, soil texture and soil minerals (all $P > 0.05$; Supplementary Fig. 8). These comparisons illustrated that permafrost collapse did not cause soil physical mixing for the topsoil samples involved in this study, and soil layers were comparable between the collapsed and control plots.

## SOC fractionation

We separated POC and MAOC from bulk soils using a fractionation method based on a combination of density and particle size[18] using the following three steps. First, 10 g of soil was put into a 100 mL centrifuge tube, and added with 50 mL of $1.6 \, g \, cm^{-3}$ NaI. After being completely mixed, the mixture was sonicated and then centrifuged at $1800 \times g$. The floating particulate organic matter, together with the supernatant, was poured into a GF/C filter membrane for filtration, completely washed with deionized water, and then dried at 60 °C to constant weight. Then, the C content of the particulate organic matter was determined as POC. Second, deionized water were added to the remaining soils in the tube to wash out any residual NaI. The washed soils were then separated with a 53-μm sieve. The residues on the sieve (>53 μm) were dried and determined as heavy POC. Third, the organic matter that passed through the sieve (<53 μm) was oven-dried and grinded, and the C content was determined as MAOC. The POC, heavy POC, and MAOC contents were determined using an elemental analyzer (Vario EL III, Elementar, Hanau, Germany) after these fractions were fully grinded.

We also quantified the content of OC-Fe using the citrate-bicarbonate-dithionite approach[25,28]. Specifically, each soil sample was evenly divided into two parts: one subsample was extracted using 0.27 M trisodium citrate, 0.11 M sodium bicarbonate, and 0.1 M sodium dithionite as the treatment group, and the other subsample was treated with 1.85 M sodium chloride and 0.11 M sodium bicarbonate as the control group[28]. We then measured the C content in these two subsamples with an elemental analyzer (Vario EL III, Elementar, Hanau, Germany). The difference in the C content of the soil residues between the treatment and control groups is the OC-Fe content.

## Measurements of plant and soil physicochemical properties

To examine the influence of vegetation and soil properties on soil C fractions, we measured aboveground and belowground biomass, topsoil temperature and moisture, pH, bulk density, and soil texture. Briefly, both aboveground and belowground biomass was determined by the harvesting method. All aboveground vegetation within the frame quadrat (25 × 25 cm, the number of replicates being the same as for the soil samples) was cut off and dried to constant weight at 65 °C. The aboveground biomass was determined based on the dried biomass. Roots were washed free of attached soils and separated into live and dead roots according to their color and tensile strength[73]. Live roots were then dried at 65 °C and weighed to calculate belowground biomass (the number of replicates being the same as for the aboveground biomass).

Soil temperature in the top 15 cm was determined with a digital thermometer (DS 1922L, Wdsen Electronic Technology Co., Shanghai, China), and soil moisture was measured by drying 20 g fresh soil sample at 105 °C to constant weight. Bulk density was determined by the oven-dried soil mass divided by the container volume. Soil texture was examined using a particle size analyzer (Malvern Masterizer 2000, Malvern, Worcestershire, UK) after eliminating organic matter and carbonates by utilizing hydrogen peroxide (30%) and hydrochloric acid (3 M), respectively[74,75]. Soil pH was analyzed by using a pH probe (PB-10, Sartorius, Göttingen, Germany) for a soil-water mixture (soil: water = 1:5). Soil C concentration was measured with an elemental analyzer (Vario EL III, Elementar, Hanau, Germany). Given that inorganic C was not detected in soil samples with a carbonate content

analyzer (Eijkelkamp 08.53, Eijkelkamp, Giesbeek, Netherlands), soil C was equal to SOC. It is worthy to note that biotic and abiotic parameters along the thaw sequence, such as aboveground and belowground biomass, soil temperature and moisture, pH, bulk density, soil texture, and SOC, were reanalyzed from published data in refs. 13, 68, while the parameters across the five additional sites at the regional scale were measured in this study.

## Microbial necromass C determination

We determined soil microbial necromass C, an important source of MAOC, based on the amino sugars[76]. Amino sugar analyses were conducted using the following procedure[77]. Specifically, ~0.4 g of soil was mixed with 10 mL HCl (6 M) at 105 °C for 8 h. After cooling, the hydrolysate was added with 100 μL of myo-inositol, filtered with glass fiber filters, and dried via a rotary evaporator at 52 °C. We redissolved the residues and adjusted the pH to 6.6–6.8. The precipitates were removed after centrifuging, and the supernatant was dried. Amino sugars were dissolved with methanol and centrifuged ($1000 \times g$ for 10 min) to dislodge salts. After the addition of 100 μL N-methyl-glucamine (internal standard), the residues were then transformed to aldononitrile derivatives through heating with 300 μL of derivatization reagent (hydroxylamine hydrochloride and 4-dimethylamino pyridine mixing with pyridine and methanol [4:1; v-v]) at 80 °C for 30 min. The derivatives were then acetylated through acetic anhydride (1 mL) under 80 °C for 20 min, and after cooling to room temperature dichloromethane (1.5 mL) was added. We removed excessive derivatization reagents as completely as possible by extracting them with 1 M HCl and MilliQ water. The organic phase owning to containing amino sugar derivatives was quantified by employing a gas chromatography (Agilent 6890 A, Agilent Technologies, Palo Alto, USA) coupled with a flame ionization detector and an HP-5 capillary column (25 m × 0.25 mm × 0.25 μm).

After the examination, we obtained three types of amino sugars, including muramic acid, glucosamine, and galactosamine. Bacterial necromass C was determined based on the content of muramic acid, and fungal necromass C was calculated from the content of glucosamine and muramic acid. Specifically, bacterial necromass C = muramic acid × 45, where 45 represents the conversion ratio of muramic acid to bacterial necromass C[78,79]; fungal necromass C = (mmol glucosamine – 2 × mmol muramic acid) × 179.17 × 9, among which 179.17 is the glucosamine molecular weight, and 9 indicates the conversion ratio from fungal glucosamine to fungal necromass C[78,79]. Microbial necromass C is the sum of bacterial and fungal necromass C. For a detailed description involving conversion formulas between amino sugars and microbial necromass C, see ref. 76.

## Soil mineral analysis

Given the critical role of soil minerals in protecting organic C, we measured two types of secondary minerals, including Fe oxides and phyllosilicates (known as clay minerals). For Fe oxides, we analyzed the amount of pedogenic Fe oxides, poorly crystalline Fe oxides, and organically complexed Fe oxides. Briefly, pedogenic Fe oxides were extracted using the citrate-bicarbonate-dithionite method[44], and poorly crystalline Fe oxides were extracted with acid-ammonium oxalate[19]. Sodium pyrophosphate was used to extract organically complexed Fe oxides[48]. The concentration of elements in the solution was examined using an inductively coupled plasma optical emission spectrometer (iCAP 6300, Thermo Fisher Scientific, Waltham, USA).

With respect to phyllosilicates, we employed X-ray diffraction (XRD) analysis to identify clay-sized minerals (<2 μm) following the modified method described by ref. 80. (i) Organic matter and carbonates were removed by hydrogen peroxide (30%) and acetic acid (1 M), respectively. These samples were then washed with deionized water. Based on Stoke's law, the clay fraction was isolated through

sedimentation[81]. (ii) We prepared three oriented clay specimens for each sample by air-drying glass slide mounts coated with clay suspension. The first one was an air-dried clay specimen, the second clay slide was saturated with ethylene glycol, and the third was treated with a combination of ethylene glycol saturation and heating to 550 °C[80]. (iii) These three clay slides were scanned between 3 and 30° (2$\theta$) with a 0.02° (2$\theta$) increment using a D8 Advance X-ray diffractometer (Bruker, Karlsruhe, Germany)[80,81].

Additionally, we examined the specific surface area of mineral-associated organic matter. Before the analysis, we removed organic matter by hypochlorite oxidation (five times with 1 M NaClO)[82]. After that, these treated samples were first degassed with helium at 325 °C for 4 h[83]. Then, nitrogen was dosed on the surfaces at −196 °C under a partial pressure (P/P$_0$) ranging from 0.04 to 0.30 in a surface area and porosity analyzer (BSD-PM2, Beishide Instrument, Beijing, China). The specific surface area was calculated based on the multipoint Brunauer-Emmett-Teller approach[84].

## Statistical analyses

All data were tested and transformed to ensure the normality of variance, and then analyzed with the following three steps. First, we used one-way ANOVAs with Least Significant Difference (LSD) multiple comparisons to analyze the differences in SOC and its associated fractions (POC, heavy POC, MAOC, and OC-Fe) as well as biotic (aboveground and belowground biomass, microbial residual C) and abiotic (soil temperature and moisture, pH, texture, Fe oxides, clay minerals, and specific surface area) variables along the thaw sequence. In these analyses, permafrost thaw stages (non-collapse, collapsed for 1 year, 10 years, and 16 years) were treated as the between-subject effect. Based on the same statistical methods, we also analyzed the differences in soil C fractions (or biotic and abiotic variables) between collapsed and control plots at each of the regional thermokarst-impacted sites.

Second, we performed ordinary least squares regression analyses to explore the relationships of POC and OC-Fe with potential factors, including vegetation parameters (aboveground and belowground biomass), soil properties (soil temperature and moisture, pH, and texture), and Fe minerals (pedogenic Fe oxides, poorly crystalline Fe oxides, and organically complexed Fe oxides) along the thaw sequence. We then conducted a random forest model to explore the relative importance of explanatory variables in affecting POC and OC-Fe using the randomForest package.

Third, we performed linear regression analyses to assess the relationships of response ratio (RR, the ratio of two variables between collapsed and control plots [Variable$_{collapse}$/Variable$_{control}$]) of POC and OC-Fe with RR of explanatory variables across the thermokarst-impacted sites at the regional scale. Statistical differences were considered to be significant at the level of $P < 0.05$. All the statistical analyses were performed in R 4.0.4 (R Core Team, 2020).

## Data availability

All data used in this study are included in supplementary information (Supplementary Data 1), and could also be downloaded from the figshare database (https://doi.org/10.6084/m9.figshare.20154044).

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

## Acknowledgements

This work was supported by the National Natural Science Foundation of China (31988102, 31825006, 91837312, and 32101332), the Second Tibetan Plateau Scientific Expedition and Research (STEP) program (2019QZKK0106 and 2019QZKK0302), and the Fundamental Research Foundation of Chinese Academy of Forestry (CAFYBB2020MA008).

## Author contributions

Y.Y. and F.L. designed the study. F.L. performed the field sample collection. F.L., S.Q., and K.F. conducted the experiments. F.L. analyzed the data. F.L., Y.Y., L.C., S.Q., Y.P., and P.S. wrote the manuscript.

## Competing interests

The authors declare no competing interests.
