## [Peer Review File · Nature Communications]

Divergent changes in particulate and mineral-associated organic carbon upon permafrost thawREVIEWER COMMENTS

Reviewer #1 (Remarks to the Author):

This manuscript by Liu and colleagues explores the fate of different carbon pools following permafrost thaw across an impressive range of sites in the Tibetan plateau. They convincingly demonstrate that mineral-associated organic carbon (MAOC) is enriched following thaw, whilst particulate organic carbon is increasingly lost with time since thaw. Importantly they also demonstrate that POC loss is associated with enhanced post-thaw microbial decomposition whilst MAOC content is controlled by both mineral protection and microbial C inputs.

This work is timely and important, coming at a time where there is significant discussion surrounding the fate of permafrost carbon and a growing appreciation of the role of MAOC in permafrost regions. It is not the first to look at MAOC in these types of systems (or indeed in permafrost systems in the Tibetan plateau) but I think makes an important contribution to the existing literature by looking at multiple sites and by identifying potential drivers that alter these pools.

Some clarification on the methodology and interpretations are required as outline below, but overall the conclusions are supported by the evidence and the methodology is fairly standard in the field. I would however have benefitted from more elaboration on the sampling scheme, which was somewhat vague (outlined in specific comments below).

One thing I would also have liked to see clarified further is whether the increase in the proportion of MAOC is purely the result of POC depletion over time, or whether the absolute content of MAOC increases post-thaw. This has important implications for explaining why MAOC increases and I feel warranted further elaboration.

Although the writing suffers from some grammatical issues, I found the arguments to be logical, well-presented and, overall, an enjoyable read.

My general suggestions would thus be to:

1. Clarify in result and discussion whether the increase in the proportion of MAOC is purely the result of POC depletion over time, or whether the absolute content of MAOC increases post-thaw.

2. Increase discussion of results within context of existing papers on Fe-OC in permafrost regions e.g. the papers from Mu et al:

a. <https://agupubs.onlinelibrary.wiley.com/doi/full/10.1002/2016GL070071>

b. <https://www.sciencedirect.com/science/article/pii/S0341816219304242>

3. Clarify methodology according to the points below

My specific comments can be found below:

62-63: Can the authors please clarify what is meant by “sign and magnitude”?

114: compared with “the” non-collapsed control

133-134: Can the authors clarify whether the increase in AGB and BGB after 10 years is referring to an increase in all of the study sites, or only in the SLH thaw sequence?

148: Suggest replacing enhanced with “increased”

150-153: Since the OC-Fe quantification from dithionite extraction is the only direct measurement of the content of mineral-associated carbon it would be nice to see these data given a little more discussion time in the main text.

152-155: Can the authors clarify here whether they mean the absolute abundance of MAOC increased or whether the percentage of MAOC as a fraction of the total increased?

171: Can the authors clarify what “responsible variable” they refer to here?

198: I’m not sure if the authors really mean by “size of microorganisms here” as I don’t see a reason why soil moisture would make an individual microbe bigger or smaller. Might just be a word choice issue.

245-241: It would be really helpful to clarify whether MAOC increases because it simply is not decomposed as easily or whether new MAOC is forming after thaw

321-326: I found this description of the sampling scheme difficult to visualise from the description. Could the authors add a summary or schematic in the SI to elaborate on the distance between these plots, how many there were and how many replicate samples were collected within them?

321-326: Was surface vegetation removed prior to sampling?

326: "In total we acquired" instead of "we totally required"

337-338: I'm not sure I quite follow what the sampling protocol is here. Did the authors collected 5 samples from different parts of the quadrant and mix them together 5 times? Or are their 5 replicates the samples taken from the different parts of the quadrat?

368: Can the authors clarify what is meant by "floating on solution"?

383: Suggest revisiting sentence structure in the line as it was difficult to follow: "Then pick out the dead roots in topsoil (drilled by 3 cm diameter corer) according to root colour and tensile strength, dry the live roots to constant weight to calculate BGB (the replicates same to AGB)."

391: I'm not sure what the meaning or relevance of "(even less in this research soil samples)"

405: think it should be "followed by extraction with 0.5M K₂SO₄."

408: divided by

Line 408: Can the authors elaborate on where this conversion value comes from?

430: using the following procedure

444: owing to containing

470-472: Can the authors clarify whether there was any replication in this DCB process? What was the concentration of dithionite and NaCl used?

Reviewer #2 (Remarks to the Author):

I fully agree with the authors, it is needed to get into a more detailed analyses of the specific pools and fractions that determine carbon storage in permafrost affected soils. Thus, the authors provide an important study that combines the analysis of specific soil C fractions with a temporal development of these as affected by climate change. The work has thus the potential to provide insights in the alteration of soil carbon storage induced by permafrost collapse, an important information for future soil carbon and ecosystem modelling.

Although the work is of great importance there are several points that need to be addressed. The used space for time approach needs further support in terms of soil textural and mineralogical data to demonstrate a high level of comparability between the sites. The changes in POM vs. MAOM have to be re-assessed with respect to demonstrating that these are not just relative but also absolute shifts in the storage. This goes along with a clear demonstration and discussion that altered POM / MAOM ratios are not due to differences in the studied soil horizons as affected by physical mixing and translocation due to thaw phenomena at the collapsed sites. These points are also highlighted in the specific remarks below:

line 76-80 Why would be an enhanced mineral association of OC be bad in this respect? If climatic stabilization vanishes, MAOM might be one key pool for persistent OC.

line 85 thermokarst comes in a lot of forms, thus it is too reductionist in this respect, please specify.

line 87-88 It is more the other way around, OM sorbes to mineral surfaces.

line 94-97 How comparable are these sites in terms of soil type, parent material, elevation, vegetation etc.? Provide data to support this assumption.

line 134 - 136 Please give explanation for the site names / at least name them in the figure caption. In the text itself it always hampers the reading flow having such abbreviations, thus try to avoid them.

line 138 Please give information on soil texture and mineralogy (clay minerals) of the studied soils to demonstrate comparability.

line 142-144 Thus, as there were clear differences in the texture between the soils, how do POM / MAOM ratios relate between them? Texture is a clear driver for the fate of POM vs. MAOM, thus it has to be discussed how textural differences interfere with the overall shifts, and or control contents.

line 153 OM-Fe assemblages are only one means of forming MAOM, thus it only represents a part of the whole MAOM pool.

line 154-155 Mainly indicating that in drier soils the amount of FeIII oxides is higher and thus there is more OC association with more stable forms of FeO (see Patzner et al. Nat Comms. 2020).

line 163-165 Soil enzymes are more or less an indicator of stoichiometric needs of microbiota, so it's questionable if this is really causation. Whereas, soil moisture directly links with microbial activity and thus the bioavailability of OM, especially POM.

line 169-170 These are not "regulating" MAOC. OC-Fe indicates just that Fe is important for OM storage in the MAOM fraction, it is not "regulating" it. And microbial necromass is one major contributor to MAOM (see work on microbial residues as precursors for MAOM). Thus, if MAOM is high, also microbial necromass should be high and vice versa.

line 192-195 How comparable are the soil layers / horizons that were studied? Did thaw processes lead to an increased admixing of top and subsoils and thus a dilution of POM?

line 212-217 POC is not necessarily more accessible than MAOC. There are for instance occluded POM fractions that are known to be rather inaccessible to microbiota due to their spatial arrangement in the soil. However, the less decomposed POM is, which means the higher the plant derived OM content, the better bioavailable is the POM. Thus, it is basically the chemical composition of the POM that drives its decomposition. An indication for that in your data is the C/N ratio.

line 219-221 Although enzymes can be long lived, these values present a short time information about the microbial OC consumption. The POC to MAOC ratio is the result of longer lasting decomposition processes. Thus, enzyme activity is more an indicator of the current environmental (e.g. aerobic vs. anaerobic conditions) and nutrient availability (stoichiometry) conditions in the respective soils.

line 227-229 These environmental factors triggered specific shifts in microbial OM consumption which lead to differences in enzyme production. Enzymes per se are not active entities that react to for instance altered accessibility of OM or stoichiometric needs of microbiota.

line 232 Although it was reported before, assumed from the recovery of occluded POM, aggregation per se might not be a key process in these soils at initial stages. Aggregation might play a larger role in more degraded mineral rich soils after permafrost thaw.

line 241-243 The main driver for the increased decomposition of especially POM rich OC pools is, the higher aeration, and higher temperatures. One can assume that the POM is not yet more stabilized due to aggregation in these permafrost affected soils.

line 245-248 Please clearly show that there is an actual increase in MAOC and not just a relative increase due to decreasing amounts of POC (see data in supplement). As the MAOC C contents are decreasing with permafrost collapse it seems there is a deepening of the profiles that leads to an admixing with minerals that are lower in OC.

line 270 by Fe/Al oxides, but especially also by clay minerals!

line 270-272 This sentence is hard to understand, please rephrase.

line 273-274 What has this to do with your data? Yes, mineral surfaces have charged surfaces that can be act as sites for OM sorption. But as you have no information about specific surface area or other mineralogical data this is just speculative.

line 274-276 This is mixing up different concepts. Plant roots and hyphae especially foster macroaggregation (see literature for root/fungal effects on macroaggregation), whereas MAOM is enriched in microaggregates.

line 276-279 Again, is this a total increase or just due to relative effects of lower POM and changes in admixing with different soil material?

line 358-361 With such an active layer depth I would assume that the soils rather classify as Cryosols given the massive permafrost layer within the profile depth.

line 370-371 How were the filtered particles prepared for C analysis. Was also N determined? If so, please report C/N ratios for all OM fractions (see also comments above).

line 390-391 Please explain how OM and carbonates were removed. Please give texture data for all soils, not only the ones normalized by OC.

Supplement Table 2, The changes in the C content of the POM look like there was a change in admixing of minerals in the POM separates. When fractionating a clean POM the composition and thus the C content should stay approx comparable. It would be good to check the C/N ratios to better describe what lead to this change in C contents.

Responses to Reviewer #1

[Comment 1] This manuscript by Liu and colleagues explores the fate of different carbon pools following permafrost thaw across an impressive range of sites in the Tibetan plateau. They convincingly demonstrate that mineral-associated organic carbon (MAOC) is enriched following thaw, whilst particulate organic carbon is increasingly lost with time since thaw. Importantly they also demonstrate that POC loss is associated with enhanced post-thaw microbial decomposition whilst MAOC content is controlled by both mineral protection and microbial C inputs. This work is timely and important, coming at a time where there is significant discussion surrounding the fate of permafrost carbon and a growing appreciation of the role of MAOC in permafrost regions. It is not the first to look at MAOC in these types of systems (or indeed in permafrost systems in the Tibetan plateau) but I think makes an important contribution to the existing literature by looking at multiple sites and by identifying potential drivers that alter these pools.

[Response] Thanks for the reviewer's positive and insightful comments. These comments, together with those listed below enabled us to have a deeper thinking on this issue, and thus guided us to conduct a thorough revision of the original manuscript. Detailed modifications please see our responses to the following comments.

Major comments:

[Comment 2] Some clarification on the methodology and interpretations are required as outline below, but overall the conclusions are supported by the evidence and the methodology is fairly standard in the field. I would however have benefitted from more elaboration on the sampling scheme, which was somewhat vague (outlined in specific comments below).

[Response] Very good comments! Following the reviewer's comments, we have clarified our sampling scheme in the revised MS. Specifically, the sampling processes included the following two steps (Fig. R1). **At the first step, we established six collapsed plots (~15 × 10 m) along a thaw sequence** (located in Shaliuhe close to the

Qinghai Lake, Qinghai Province, China), which had collapsed for 1, 3, 7, 10, 13, and 16 years before the sampling year of 2014 (Liu et al., 2018). The collapse time in each plot was estimated by dividing the distance between the collapsed plot and the gully head by the retreat rate ($\sim 8.0 \text{ m year}^{-1}$; the rate of head-wall retreat determined by Google Earth satellite images and *in situ* monitoring) (Liu et al., 2018). Then, we set up six paired control (non-collapsed) plots adjacent to these collapsed plots. To limit experimental cost, we selected three paired control and collapsed plots (with times since collapse of 1, 10 and 16 years, representing the early, middle and late stages of collapse) to examine the responses of POC, MAOC and OC-Fe to permafrost collapse (Fig. R1). Within each collapsed plot, we collected topsoil (0-15 cm) samples from all vegetated patches (Fig. R2), and then evenly selected 10 vegetated patches for this study considering the heavy workload and high cost. In each selected vegetated patch, 5-8 soil cores were collected and completely mixed as one replicate. Within each control plot, topsoil samples were randomly collected from five quadrats at the center and four corners of the plot. In each quadrat, 15-20 soil cores were collected and mixed as one replicate. Thereby, ten replicates were acquired in each collapsed plot ($n = 10$), and five replicates were obtained in each control plot ($n = 5$). In total we acquired 45 soil samples, including 30 samples from three collapsed plots and 15 samples from the non-collapsed control for subsequent analysis.

At the second step, to further verify the universality of collapse effects on SOC fractions, **we collected topsoil (0-15 cm) samples from an additional five similar sites located near the towns of Ebo, Mole, Huashixia and Huanghe across a 550-km permafrost transect in August 2020 (Fig. R1).** Specifically, paired collapsed and control plots ($15 \times 10 \text{ m}$) were established at the end of a gully and in adjacent non-collapsed areas in each site. In the collapsed plot, we set five $5 \times 3 \text{ m}$ quadrats at the center and four corners of the plot, and then collected topsoil samples within all patches in these quadrats. In each quadrat, all collected soil cores (15-20 cores) were completely mixed as one replicate, and finally five replicates were obtained in each plot ($n = 5$).

Similarly, five replicates were obtained from the five quadrats in each control plot ($n = 5$). In total, we collected 50 topsoil samples across these five thermokarst-impacted sites. We have rephrased the sampling method in the revised MS (Pages 16-17, line 331-368) and added a schematic diagram in the supplementary materials (Fig. R1; Pages 7-8, line 44-58 in the supplementary materials).

Step 1: Sampling along the thaw sequence

Step 2: Sampling across the regional sites

Fig. R1. Schematic diagram of soil sampling along the thaw sequence (SLH) and across the regional sites (marked as FEB, SEB, ML, HSX and HH) on the Tibetan Plateau. a, Location of the permafrost thaw sequence based on the map of permafrost distribution on the Tibetan Plateau (Zou et al., 2017). b, Image of the thermo-erosion gully captured from coloured LiDAR point cloud data (VZ-400, Riegl, Horn, Austria) with the specific plot distribution along the thaw sequence. Control #1, Control #2 and Control #3 represent three non-collapsed plots which were paired to collapsed plots occurring for 1 year, 10 years and 16 years, respectively. c, Sampling schematic diagram for the thaw sequence (photo credit: F.T. Liu). d, Distribution of five thermokarst-impacted sites at the regional scale. e, Landscapes of the regional sites with non-collapsed control and collapsed plots (photo credit: Z.L. Li). f, Sampling schematic diagram for the regional sites. SLH, site at Shaliuhe; FEB, the first site at Ebo; SEB, the second site at Ebo; ML, site at Mole; HSX, site at Huashixia; HH, site at Huanghe. Notably, along the thaw sequence, the distance between the collapsed plots occurring for 1 year and those for 10 and 16 years is 80 m and 130 m respectively, while the distance between the collapsed and non-collapsed control plots is less than 1 m. Across the five sites over the regional scale, we set up one paired collapsed and control plots (15 × 10 m) at the end of a gully and in adjacent non-collapsed areas at each site. These five sites were distributed across a 550-km permafrost transect on the Tibetan Plateau. Under our sampling design, there were five replicates in the control plot and ten replicates in the collapsed plots along the thaw sequence. Across the five thermokarst-impact sites over the regional scale, there were five replicates in both control and collapsed plots.

Fig. R2. Picture showing vegetated patches (marked with white border) and exposed patches (marked with yellow border) within the collapsed plot across the Tibetan thermokarst-impacted sites (photo credit: F.T. Liu). The inset shows topsoil sampling in vegetated patches. Notably, to avoid the interference of soil layer mixture, we collected topsoil (0-15 cm) within the vegetated patches (40-60 cm thickness) rather than within the exposed patches, in which soil cores were at least 10 cm away from the edge of vegetated patch.

[Comment 3] One thing I would also have liked to see clarified further is whether the increase in the proportion of MAOC is purely the result of POC depletion over time, or whether the absolute content of MAOC increases post-thaw. This has important implications for explaining why MAOC increases and I feel warranted further elaboration.

[Response] Very good comment! To address this comment, we have calculated the absolute content of MAOC, and also analysed the correlations between the proportions of MAOC and POC in the revised MS. **Our results revealed that the absolute content of MAOC showed no significant changes after permafrost collapse, both along the**

permafrost thaw sequence and at four of the five thermokarst-impact sites over the regional scale (Fig. R3). We also found that the proportion of POC to SOC was negatively correlated with that of MAOC (Fig. R4). Based on these two points, we agree with the reviewer’s point that the increased proportion of MAOC could be due to the decrease of POC. Therefore, we have added the data and related description involving the absolute contents of POC and MAOC in the revised MS, **discussed why POC content decreased but MAOC content kept relatively stable upon permafrost thaw, and further changed our original argument from “permafrost collapse could result in MAOC enrichment” to “Divergent changes in particulate and mineral-associated organic carbon upon permafrost thaw”**. We have clearly stated this point in the revised MS (Page 1, line 1-2; Pages 6-7, line 109-133; Pages 11-12, line 226-258; Pages 14-15, line 293-307). Thanks for your understanding!

Fig. R3. Changes in the absolute contents of POC and MAOC induced by thermokarst formation. a-b, Variations in contents of POC (a) and MAOC (b) along the thaw sequence. c-d, Comparisons of POC (c) and MOAC (d) contents between

collapsed (dark colour) and control (light colour) plots at each of the five regional thermokarst-impacted sites: FEB, SEB, ML, HSX and HH. POC, particulate organic carbon; MAOC, mineral-associated organic carbon. FEB, the first site at Ebo, SEB, the second site located at Ebo. ML, HSX and HH indicate the sites at the Mole, Huashixia and Huanghe, respectively. Different capital letters indicate significant differences for the variables within plots along the thaw sequence (LSD test, $P < 0.05$). Dashed lines distinguish different thermokarst-impacted sites, denoting that the parameters are compared between collapsed and control plots in each site rather than across various study sites. * $P < 0.05$ and ** $P < 0.01$.

Fig. R4. Correlations between the percentages of POC and MAOC along the permafrost thaw sequence (a) and across the five sites over the regional scale (b). FEB, the first site at Ebo, SEB, the second site at Ebo. ML, HSX and HH indicate the sites at Mole, Huashixia and Huanghe, respectively. * $P < 0.05$ and ** $P < 0.01$.

[Comment 4] Although the writing suffers from some grammatical issues, I found the arguments to be logical, well-presented and, overall, an enjoyable read.

[Response] Following the reviewer's comments, we have corrected the grammatical issues in the revised MS, and also invited two native English speakers (**Dr. Alistair**

Culf and Prof. Pete Smith) and an English language editing service (*i.e.*, **Springer Nature Author Services**) for language check throughout the main text. Please see the certification at the end of this response letter.

[Comment 5] My general suggestions would thus be to: 1. Clarify in result and discussion whether the increase in the proportion of MAOC is purely the result of POC depletion over time, or whether the absolute content of MAOC increases post-thaw.

[Response] Following the reviewer's comments, we have clarified that the absolute MAOC content remained stable while the proportion of MAOC to SOC significantly increased along the thaw sequence and at four of the five thermokarst-impacted sites. Our results revealed that the increased proportion of MAOC could be associated with the decrease of POC. We have clearly stated this point in the *Results* and *Discussion* sections of the revised MS, and **discussed why POC content decreased but MAOC content kept relatively stable upon permafrost thaw** (Pages 6-7, line 109-133; Pages 11-12, line 226-258; Pages 14-15, line 293-307).

[Comment 6] 2. Increase discussion of results within context of existing papers on Fe-OC in permafrost regions e.g. the papers from Mu et al:

a. <https://agupubs.onlinelibrary.wiley.com/doi/full/10.1002/2016GL070071>

b. <https://www.sciencedirect.com/science/article/pii/S0341816219304242>

[Response] Following the reviewer's suggestions, **we have added the related discussion concerning OC-Fe and cited the papers by Mu et al. in the revised MS as follows:** *“In support of our third hypothesis, both the OC-Fe content and its proportion to SOC significantly increased after permafrost collapse, both along the permafrost thaw sequence and at most of the thermokarst-impacted sites over the regional scale. Specifically, the amount of OC-Fe and its proportion to SOC increased significantly by 21.9% and 67.5% after permafrost collapse, respectively (Figs. 2 and 3). Our findings of enriched OC-Fe after permafrost collapse was supported by earlier observations across the study area (Mu et al., 2016, 2020), which jointly*

illustrated widespread accumulation in OC-Fe after upland thermokarst formation.

Interestingly, this enrichment in OC-Fe after upland thermokarst distinctly differs from the observations from lowland thermokarst in the Arctic, which have revealed reduced OC-Fe upon permafrost thaw (Patzner et al., 2020). Such a discrepancy could be largely due to the different trajectories of soil moisture after thermokarst formation (Huang and Hall, 2017; Mu et al., 2020; Joss et al., 2022; Patzner et al., 2022), which acts as a critical driver for Fe(III) reduction and Fe(II) oxidation (Patzner et al., 2020; Monhonval et al., 2021; Joss et al., 2022; Patzner et al., 2022). Specifically, water-logging is generally induced in lowland regions (Hodgkins et al., 2014; McCalley et al., 2014; Patzner et al., 2020), while soil drainage often occurs in upland areas upon permafrost collapse (Schuur et al., 2008; Abbott and Jones, 2015; Liu et al., 2018). It has been demonstrated that water-logging favours the reductive dissolution of reactive Fe(III) minerals (also referred to as Fe oxides) because Fe(III)-bearing minerals can be used as electron acceptors for anaerobic respiration by Fe(III)-reducing microorganisms (Chen et al., 2020; Patzner et al., 2020; Patzner et al., 2022). Given that reactive Fe(III) minerals could stabilize SOC via sorption/co-precipitation and protect it from microbial degradation (Wiesmeier et al., 2019; Chen et al., 2020; Kleber et al., 2021), the Fe(III) reduction occurring in lowland thermokarst could result in the loss of OC-Fe. Conversely, in upland thermokarst, with the improvement of aeration after thermokarst formation, oxygen can replace Fe(III) as electron acceptors for microbial respiration, and also stimulate the Fe(II) oxidation to Fe(III), thereby promoting the formation of OC-Fe associations (Chen et al., 2020; Patzner et al., 2022). In support of this deduction, Fe oxides were observed to increase significantly after thermokarst formation across our study sites (Supplementary Fig. 1g-i). Moreover, both Fe oxides and OC-Fe were negatively correlated with soil moisture (Supplementary Fig. 4), which jointly demonstrated that soil drainage after upland thermokarst formation was the main reason for the accumulation of Fe oxides and OC-Fe. Consequently, the improvement of soil aeration induced by permafrost collapse in upland regions could elevate the content of Fe oxides and enhance the protective effects of Fe phases on soil

C pools, ultimately resulting in the accumulation of OC-Fe” (Pages 13-14, line 260-291).

[Comment 7] 3. Clarify methodology according to the points below

[Response] Following the reviewer’s suggestion, we have drawn a schematic diagram in the supplementary information (**Fig. R1**) and re-organized the descriptions of *Method* section to clarify our site design and sampling methodology. In brief, our sampling procedure included the following two steps. **At the first step**, we established six paired collapsed and control plots along the thaw sequence, and then selected three paired plots (representing sites with non-collapse and collapsing for 1, 10 and 16 years) to examine the responses of POC, MAOC and OC-Fe to permafrost collapse. **At the second step**, we collected additional soil samples from five similar sites (FEB, SEB, ML, HSX and HH) across the Tibetan Plateau to further verify the universality of observations along this thaw sequence. The specific sampling methodology has been clearly described in the *Methods* section of the revised MS (Pages 16-17, line 331-368).

Minor comments:

[Comment 8] My specific comments can be found below: 62-63: Can the authors please clarify what is meant by “sign and magnitude”?

[Response] The “sign and magnitude” means the “direction (increase or decrease) and magnitude of soil organic C changes under various climate change scenarios”. To avoid the confusion, this sentence has been modified as follows: “*Site-level observations have reported that permafrost thaw could lead to a large amount of soil C loss within years and decades (Pizano et al., 2014; Abbott and Jones, 2015; Liu et al., 2018), but substantial uncertainties exist in the model-projected **direction and magnitude** of soil organic C (SOC) changes under various climate change scenarios (Todd-Brown et al., 2014; McGuire et al., 2018)*” (Page 3, line 57-60).

[Comment 9] 114: compared with “the” non-collapsed control

[Response] Done as suggested (Page 6, line 108).

[Comment 10] 133-134: Can the authors clarify whether the increase in AGB and BGB after 10 years is referring to an increase in all of the study sites, or only in the SLH thaw sequence?

[Response] The increase in AGB and BGB after 10 years is only referring to an increase along the SLH thaw sequence, not in all of the study sites. To clearly state this point, we have reorganized this section as follows: “Specifically, along the thaw sequence located at Shaliuhe, **both aboveground and belowground biomass significantly increased in the 10 years since permafrost collapse** (all $P < 0.05$; Supplementary Fig. 1a-b), although there were no significant changes in the early or late stages of permafrost collapse (collapsed for 1 year and 16 years, respectively; all $P > 0.05$). **Four of the five additional sites at the regional scale, those at Ebo, Mole, and Huashixia, showed no obvious changes in either aboveground or belowground biomass after permafrost collapse** (all $P > 0.05$; Supplementary Fig. 1a-b). However, at the Huanghe site, belowground biomass significantly increased ($P < 0.05$; Supplementary Fig. 1b) while aboveground biomass did not show any significant variations ($P > 0.05$; Supplementary Fig. 1a)” (Pages 7-8, line 143-152).

[Comment 11] 148: Suggest replacing enhanced with “increased”

[Response] Done as suggested (Page 8, line 163).

[Comment 12] 150-153: Since the OC-Fe quantification from dithionite extraction is the only direct measurement of the content of mineral-associated carbon it would be nice to see these data given a little more discussion time in the main text.

[Response] Following the reviewer’s suggestion, **we have added a new paragraph to discuss OC-Fe in the revised MS as follows**: “In support of our third hypothesis, both the OC-Fe content and its proportion to SOC significantly increased after permafrost collapse, both along the permafrost thaw sequence and at most of the

thermokarst-impacted sites over the regional scale. Specifically, the amount of OC-Fe and its proportion to SOC increased significantly by 21.9% and 67.5% after permafrost collapse, respectively (Figs. 2 and 3). Our findings of enriched OC-Fe after permafrost collapse was supported by earlier observations across the study area (Mu et al., 2016, 2020), which jointly illustrated widespread accumulation in OC-Fe after upland thermokarst formation. Interestingly, this enrichment in OC-Fe after upland thermokarst distinctly differs from the observations from lowland thermokarst in the Arctic, which have revealed reduced OC-Fe upon permafrost thaw (Patzner et al., 2020). Such a discrepancy could be largely due to the different trajectories of soil moisture after thermokarst formation (Huang and Hall, 2017; Mu et al., 2020; Joss et al., 2022; Patzner et al., 2022), which acts as a critical driver for Fe(III) reduction and Fe(II) oxidation (Patzner et al., 2020; Monhonval et al., 2021; Joss et al., 2022; Patzner et al., 2022). Specifically, water-logging is generally induced in lowland regions (Hodgkins et al., 2014; McCalley et al., 2014; Patzner et al., 2020), while soil drainage often occurs in upland areas upon permafrost collapse (Schuur et al., 2008; Abbott and Jones, 2015; Liu et al., 2018). It has been demonstrated that water-logging favours the reductive dissolution of reactive Fe(III) minerals (also referred to as Fe oxides) because Fe(III)-bearing minerals can be used as electron acceptors for anaerobic respiration by Fe(III)-reducing microorganisms (Chen et al., 2020; Patzner et al., 2020; Patzner et al., 2022). Given that reactive Fe(III) minerals could stabilize SOC via sorption/co-precipitation and protect it from microbial degradation (Wiesmeier et al., 2019; Chen et al., 2020; Kleber et al., 2021), the Fe(III) reduction occurring in lowland thermokarst could result in the loss of OC-Fe. Conversely, in upland thermokarst, with the improvement of aeration after thermokarst formation, oxygen can replace Fe(III) as electron acceptors for microbial respiration, and also stimulate the Fe(II) oxidation to Fe(III), thereby promoting the formation of OC-Fe associations (Chen et al., 2020; Patzner et al., 2022). In support of this deduction, Fe oxides were observed to increase significantly after thermokarst formation across our study sites (Supplementary Fig. 1g-i). Moreover, both Fe oxides and OC-Fe were

negatively correlated with soil moisture (Supplementary Fig. 4), which jointly demonstrated that soil drainage after upland thermokarst formation was the main reason for the accumulation of Fe oxides and OC-Fe. Consequently, the improvement of soil aeration induced by permafrost collapse in upland regions could elevate the content of Fe oxides and enhance the protective effects of Fe phases on soil C pools, ultimately resulting in the accumulation of OC-Fe” (Pages 13-14, line 260-291).

[Comment 13] 152-155: Can the authors clarify here whether they mean the absolute abundance of MAOC increased or whether the percentage of MAOC as a fraction of the total increased?

[Response] Yes, this sentence in the original MS means that the percentage of MAOC increased rather than that the absolute abundance of MAOC increased. Considering the current and previous comments ([Comments 3 and 5]), **we not only focused on the increased percentage of MAOC to SOC, but also emphasized the stable MAOC content after permafrost collapse in the revised MS.** Hence, **we have rephrased these sentences as follows:** “By comparison, the MAOC content did not exhibit significant changes before and after permafrost collapse at four of the five sampling sites (all $P > 0.05$; Fig. 2i). Despite the relatively stable MAOC, the OC-Fe content significantly increased after permafrost collapse, both along the thaw sequence and at four of the five regional-scale sites (all $P < 0.05$; Fig. 2e, j). These results illustrated that SOC loss was largely driven by the decrease of POC and heavy POC, while MAOC remained stable and OC-Fe continuously accumulated” (Page 6, line 117-123).

[Comment 14] 171: Can the authors clarify what “responsible variable” they refer to here?

[Response] The “responsible variable” referred to “OC-Fe” in the original MS. To avoid the confusion, **we have rephrased this sentence as follows:** “The organically complexed Fe oxides were positively correlated with **OC-Fe** ($P < 0.001$; Fig. 4e), indicating that the increased Fe oxides and improved soil aeration induced by

permafrost collapse could result in the accumulation of OC-Fe along the thaw sequence”
(Page 9, line 187-190).

[Comment 15] 198: I’m not sure if the authors really mean by “size of microorganisms here” as I don’t see a reason why soil moisture would make an individual microbe bigger or smaller. Might just be a word choice issue.

[Response] Sorry for the inaccurate description. Combining the suggestions from this reviewer and Reviewer#2, the paragraph has been reorganized and this sentence has been deleted in the revised MS. Thanks for your understanding!

[Comment 16] 245-241: It would be really helpful to clarify whether MAOC increases because it simply is not decomposed as easily or whether new MAOC is forming after thaw

[Response] Very good point! After analysing the absolute content of MAOC, we found that **the absolute MAOC content did not exhibit significant increase, but remained relatively stable before and after permafrost collapse** (Fig. R3b, d). The stable MAOC could be due to the fact that it simply is not decomposed as easily. Specifically, MAOC is protected by Fe oxides and clay minerals through sorptive interactions (Rowley et al., 2018; Lavalley et al., 2020; Lugato et al., 2021; Cotrufo and Lavalley, 2022). In particular, clay-sized particles including Fe oxides and phyllosilicates can provide a high specific surface area, and thus adsorb substantial amounts of SOC (von Lützow et al., 2006; Wiesmeier et al., 2019; Chen et al., 2020). These organo-mineral complexes could stabilize soil C through the formation of chemical bonds, and subsequently shield SOC against microbial decomposition (von Lützow et al., 2006; Gentsch et al., 2018; Fulton - Smith and Cotrufo, 2019; Wiesmeier et al., 2019; Chen et al., 2020). Consequently, the higher degree of mineral protection lead to the stable MAOC content (Marin-Spiotta et al., 2009; Gentsch et al., 2015; Karhu et al., 2019; Lavalley et al., 2020).

Another possible reason for the stable MAOC is that the loss of extant MAOC may be offset by the newly generated MAOC. Although MAOC is protected by soil minerals, it could still be partly decomposed or laterally transferred over long-term scales (*i.e.*, decadal or centennial timescales) (Hall et al., 2015; Stacy et al., 2019). However, this part of MAOC loss might be offset by the increased microbial residual C (an important source of MAOC) after permafrost collapse (Fig. R5), and thus results in relatively stable MAOC content (Cotrufo et al., 2013, 2022; Lavallee et al., 2020). To better understand the potential mechanisms of MAOC dynamics, **stable isotope labelling experiment (¹³C labelled plant material) should be used to trace the formation and loss of MAOC** (Cotrufo et al., 2015, 2022; Lavallee et al., 2018; Sokol and Bradford, 2018). Nevertheless, such an operation is difficult to be carried out *in situ* given that it would need a relatively long time to trace the carbon fate because either the transformation from plant carbon to MAOC or its decomposition acquires long time (Cotrufo et al., 2015; Sokol and Bradford, 2018; Sokol et al., 2019). **We have clearly mentioned this point in the Discussion section of the revised MS, and highlighted that future studies should distinct whether the variations of MAOC is caused by the balance of carbon input and output based on an isotope labelling experiment** (Page 12, line 238-244 and line 249-258).

Fig. R5. Changes in microbial necromass carbon content and its proportion to bulk soil carbon induced by thermokarst formation. a-b, Changes in microbial necromass carbon content along the thaw sequence (a) and at the five additional sites over the regional scale (b). c-d, Shifts in the proportion of microbial necromass carbon to bulk soil carbon along the thaw sequence (c) and at the regional thermokarst-impacted sites (d). FEB, the first site at Ebo; SEB, the second site at Ebo; ML, site at Mole; HSX, site at Huashixia; HH, site at Huanghe. Error bars represent standard errors. Different capital letters indicate significant differences for the variables within plots along the thaw sequence (LSD test, $P < 0.05$). Dashed lines distinguish different thermokarst-impacted sites, denoting that the parameters are compared between collapsed and control plots in each site rather than across various study sites. * $P < 0.05$, ** $P < 0.01$, and *** $P < 0.001$.

[Comment 17] 321-326: I found this description of the sampling scheme difficult to visualise from the description. Could the authors add a summary or schematic in the SI to elaborate on the distance between these plots, how many there were and how many replicate samples were collected within them?

[Response] Following the reviewer's comment, we have drawn a schematic (Fig. R1) and added it to the supplementary materials of the revised MS (Pages 7-8, line 44-58 in the supplementary materials) to elaborate the distance between these plots, how many there were and how many replicate samples were collected in this study. Specifically, we set up three paired collapsed and control plots along the thaw sequence, and the distance between the collapsed plots occurring for 1 year and those for 10 and 16 years is 80 m and 130 m, respectively. The distance between the collapsed and non-collapsed control plots is less than 1 m. We then acquired ten replicates in the collapsed plot ($n = 10$) and five replicates in the control plot ($n = 5$). Regarding the five sites over the regional scale, they were distributed across a 550-km permafrost transect on the Tibetan Plateau. In each site, we only established one paired collapsed and control plots (15×10 m) at the end of the gully and in adjacent non-collapsed areas. We then

collected five replicates in either collapsed or control plots ($n = 5$).

[Comment 18] 321-326: Was surface vegetation removed prior to sampling?

[Response] Surface vegetation was not removed prior to soil sampling, but was removed in the laboratory after soil sampling. Given that ~1200 soil cores from six sites should be collected in this field sampling campaign, we did not remove surface vegetation prior to soil sampling to avoid increasing workload and time in the field. Nevertheless, **we completely removed surface vegetation in the laboratory**. We have clearly described this point in the *Method* sections of the revised MS as follows: “*After transportation to the laboratory, all the soil samples were handpicked to remove surface vegetation, roots and gravels, and sieved (2 mm) for subsequent analysis*” (Page 17, line 366-368).

[Comment 19] 326: “In total we acquired” instead of “we totally required”

[Response] Done as suggested (Page 17, line 351).

[Comment 20] 337-338: I’m not sure I quite follow what the sampling protocol is here. Did the authors collected 5 samples from different parts of the quadrant and mix them together 5 times? Or are their 5 replicates the samples taken from the different parts of the quadrat?

[Response] Sorry for the unclear description. Actually, 5 replicates were taken from the five quadrats within each plot (Fig. R1). In the collapsed plot, we set five 5×3 m quadrats at the center and four corners of the plot, and then collected topsoil samples within all the vegetated patches in these quadrats. **In each quadrat, all the collected soil cores (15-20 cores) were completely mixed as one replicate, and finally five replicates were obtained from each plot ($n = 5$)**. Similarly, five replicates were obtained from the five quadrats in each control plot ($n = 5$). We have rephrased the related sentences and clearly described our sampling method in the revised MS (Page 17, line 358-365).

[Comment 21] 368: Can the authors clarify what is meant by “floating on solution”?

[Response] The “floating on solution” means the “floating particulate organic matter at solution surface”. To avoid this confusion, **we have re-organized this sentence as follows:** “*The floating particulate organic matter together with the supernatant were poured into a GF/C filter membrane for filtration, and completely washed with deionized water and then dried at 60°C to constant weight*” (Page 20, line 415-417).

[Comment 22] 383: Suggest revisiting sentence structure in the line as it was difficult to follow: “Then pick out the dead roots in topsoil (drilled by 3 cm diameter corer) according to root colour and tensile strength, dry the live roots to constant weight to calculate BGB (the replicates same to AGB).”

[Response] Following the reviewer’s comment, we have rewritten this sentence as follows: “*Roots were washed free of attached soils and separated into live and dead roots according to their colour and tensile strength (Yang et al., 2010). Live roots were then dried and weighed to calculate belowground biomass (the number of replicates being the same as for the aboveground biomass)*” (Page 21, line 442-445).

[Comment 23] 391: I’m not sure what the meaning or relevance of “(even less in this research soil samples)”

[Response] Sorry about the poor description. In the sentence of original MS, we want to describe that, even though the content of carbonates is lower across our samples, we still used the relevant method to remove them. **To avoid the confusion, we have removed this description and rephrased this sentence as follows:** “*Soil texture was examined using a particle size analyzer (Malvern Masterizer 2000, Malvern, Worcestershire, UK) after eliminating organic matter and carbonates by utilizing hydrogen peroxide (30%) and hydrochloric acid (3 M), respectively (Chen et al., 2016; Igaz et al., 2020)*” (Page 21, line 450-453).

[Comment 24] 405: think it should be “followed by extraction with 0.5M K₂SO₄.”

[Response] Considering the Reviewer#2’s comment (*[Comment 15]*), we have removed the parameter of microbial biomass C and associated method, and thus deleted this sentence in the revised MS. Thanks for your understanding!

[Comment 25] 408: divided by

[Response] We have deleted this sentence because the method involving microbial biomass C determination was removed in the revised MS.

[Comment 26] Line 408: Can the authors elaborate on where this conversion value comes from?

[Response] Before replying to this issue, we would like to mention that microbial biomass C was determined by the chloroform fumigation-extraction (FE) method (Joergensen, 1996). The basic principle of FE method is that the cell membranes of soil microorganisms are attacked by chloroform, and a part of the microbial constituents, especially in the cytoplasm, is degraded and transformed into extractable components. Then we could calculate microbial biomass C according to the following equation:

$$\text{Microbial biomass C} = E_C/k_{EC}$$

where E_C is the organic C extracted from fumigated soil minus that extracted from non-fumigated soil, and **k_{EC} is the conversion value representing the extractable part of microbial biomass C after fumigation.** Based on the experimental calculation by using 66 soils and literature data ($n = 87$) conducted by Joergensen (1996), k_{EC} of 0.45 is recommended for microbial C analysis. This k_{EC} value has also been widely used in previous studies (Vance et al., 1987; Wu et al., 1990). **Nevertheless, given that Reviewer#2 pointed out that microbial properties ([Comment 15]) were not suitable to be used for explaining the variations of POC, this indicator was removed in the revised MS.** Thanks for your understanding!

[Comment 27] 430: using the following procedure

[Response] Done as suggested (Page 22, line 466-467).

[Comment 28] 444: owning to containing

[Response] Done as suggested (Page 23, line 480).

[Comment 29] 470-472: Can the authors clarify whether there was any replication in this DCB process? What was the concentration of dithionite and NaCl used?

[Response] Yes, regarding the DCB process, there were five and ten replicates in the control and collapsed plots in the permafrost thaw sequence, respectively. Across the five sites over the regional scale, there are five replicates in both control and collapsed plots. Specifically, **each soil sample was evenly divided into two parts**: one subsample was extracted using **0.27 M trisodium citrate, 0.11 M sodium bicarbonate, and 0.1 M sodium dithionite** as the treatment group, the other subsample was treated with **1.85 M sodium chloride and 0.11 M sodium bicarbonate** as the control group (Patzner et al., 2020). We then measured the C content in these two subsamples using an elemental analyzer (Vario EL III, Elementar, Hanau, Germany). The difference in the C content of the soil residues between the treatment and control groups is the OC-Fe content. Moreover, the concentration of dithionite and sodium chloride used in this study was 0.1 M and 1.85 M respectively, which has also been widely used in previous studies (Gentsch et al., 2015; Patzner et al., 2020; Qin et al., 2021). We have clearly stated this point in the revised MS (Page 20, line 427-430)

Thanks again for the reviewer's insightful suggestions. As mentioned above, **we have redrawn a supplementary figure and rephrased the related descriptions to clarify our sampling method**. Additionally, **we have re-calculated the absolute contents of POC and MAOC and added the related discussion (including OC-Fe)**, and not only focused on the increased proportion of POC and MAOC to SOC, but also **emphasized the divergent responses of absolute contents of POC and MAOC to permafrost collapse in the revised MS** (Pages 6-7, line 109-133; Pages 11-12, line 226-258).

Particularly, we added another eight supplementary figures and three supplementary tables to support the main conclusion drawn in this study. By doing so, we feel that our conclusion becomes more convinced and the revised manuscript has been greatly improved. Thank you!

Responses to Reviewer #2

[Comment 1] I fully agree with the authors, it is needed to get into a more detailed analyses of the specific pools and fractions that determine carbon storage in permafrost affected soils. Thus, the authors provide an important study that combines the analysis of specific soil C fractions with a temporal development of these as affected by climate change. The work has thus the potential to provide insights in the alteration of soil carbon storage induced by permafrost collapse, an important information for future soil carbon and ecosystem modelling.

[Response] We are very grateful to the reviewer for the positive and insightful comments on our manuscript! These comments, together with those listed below enabled us to have a deeper thinking on this issue, and thus guided us to conduct a thorough revision of the original manuscript. Detailed modifications please see our responses to the following comments.

Major comments:

[Comment 2] Although the work is of great importance there are several points that need to be addressed. The used space for time approach needs further support in terms of soil textural and mineralogical data to demonstrate a high level of comparability between the sites. The changes in POM vs. MAOM have to be re-assessed with respect to demonstrating that these are not just relative but also absolute shifts in the storage. This goes along with a clear demonstration and discussion that altered POM / MAOM ratios are not due to differences in the studied soil horizons as affected by physical mixing and translocation due to thaw phenomena at the collapsed sites. These points are also highlighted in the specific remarks below:

[Response] Very good comments! To address the reviewers' comments, we conducted the following three major changes:

First, we added new measurements about clay minerals and specific surface area in all collected soils, which were then combined with soil texture to be used for judging

whether our study sites were similar with a high level of comparability. Before doing so, we would like to mention that **the space for time approach was only used for the site along the permafrost thaw sequence (located at Shaliuhe), not for the other five sites (located at Ebo, Mole, Huashixia and Huanghe) over the regional scale.** Along the thaw sequence, we analysed a series of parameters (*i.e.*, vegetation biomass, edaphic variables and soil minerals) for the three control plots which were located outside the gully but adjacent to three collapsed plots within the gully (Fig. R1). By comparing aboveground biomass, belowground biomass, SOC, soil moisture, pH, bulk density, soil texture and soil minerals, we observed that the above parameters were not significantly different among the three control plots along the thaw sequence (all $p > 0.05$; Table R1). **These comparisons demonstrated that the study area was homogeneous before permafrost thaw and thus it was reasonable to adopt the space for time approach along the permafrost thaw sequence.**

Regarding the other five sites over the regional scale, we only focused on the impact of permafrost collapse on POC and MAOC **by comparing soil C fractions inside and outside the gully in each site, rather than among the study sites.** Nevertheless, we also provided the information involving vegetation type, soil type, parent material, elevation, clay minerals and soil texture for our study sites in the Supplementary Figure 1 and Supplementary Table 2 according to the reviewer's suggestion (Table R2). Based on these data, even there existed some differences between study sites, the pristine soils in each site could also be regarded as homogeneous because of the low coefficient of variation of parameters in the control plot of each site (Table R3; Wiltshire et al., 1986). Given this point, these selected sites were basically in line with the experimental purpose of this study since that the parameters between collapsed and control plots were comparable and the differences in parameters inside and outside the gully could be attributed to the effects of permafrost collapse. In addition, when examining the relationships of soil C fractions with potential factors across the five sites at the regional scale, we used the response ratio (RR, $\text{Variable}_{\text{collapse}}/\text{Variable}_{\text{control}}$) of these two

variables to avoid spatial interference. Therefore, some existing differences between sampling sites over the regional scale should have limited influence on major results shown in this study. We have clearly stated these points in the revised MS (Pages 9-10, line 192-201; Page 18, line 370-388).

Second, we analysed the absolute contents of POC and MAOC, and found that both the content and the proportion of POC significantly decreased both along the permafrost thaw sequence and at the five sites over the regional scale. Regarding the variations of MAOC, **there was no significant difference in the absolute content of MAOC among most of the thermokarst-impacted sites** (Fig. R2), although the proportion of MAOC significantly increased (Fig. 3c, g). Therefore, we added data involving the absolute contents of POC and MAOC (Fig. R2), and modified the descriptions of *Results and Discussion* sections in the revised MS (Pages 6-7, line 109-133; Pages 11-12, line 226-258; Pages 14-15, line 293-307), in which **we not only focused on the proportion of POC and MAOC, but also emphasized the divergent responses of POC and MOAC contents to permafrost thaw.** Thanks for your understanding!

Third, we clarified that the collected topsoil samples were less affected by physical mixing and translocation due to thaw phenomena at the thermokarst-impacted sites. Specifically, to examine changes in soil properties upon permafrost thaw, **we chose to collect topsoil within the vegetated patches rather than from the exposed soil areas in the collapsed plots** (Fig. R3). These vegetated patches (40-60 cm thickness) are formed during the landscape fragmentation after permafrost collapse (Liu et al., 2018). Although permafrost collapse would inevitably lead to soil translocation, these vegetated patches maintained their original shapes, especially for the topsoil because it is protected by mattic epipedon in this swamp meadow on the Tibetan Plateau (which has an intensive root network protecting soils against interference; Fig. R4) (Jiang et al., 2020; Ma et al., 2020). Moreover, we collected 0-15 cm topsoil within the vegetated patches, in which soil cores were at least 10 cm away from the edge of the

patch. Due to these two points, topsoil should not be mixed with the subsoil in our case.

To test this deduction, we compared the non-collapsed (control) plot with the collapsed plot occurring for 1 year (the early stage of the permafrost thaw sequence), and observed no significant differences in soil properties such as **bulk density, SOC, pH, soil texture and soil minerals** (all $P > 0.05$; Table R4). These comparisons illustrated that permafrost collapse did not cause soil physical mixing for topsoil samples involved in this study. We have clearly stated these points in the revised MS (Pages 18-19, line 390-408).

Table R1. Comparisons of biotic and abiotic properties among three control plots which were adjacent to collapsed plots (collapsed for 1 year, 10 years and 16 years; for detailed plot distribution see Fig. R1) within the thermo-erosion gully.

Variables	Control #1	Control #2	Control #3
AGB (g m ⁻²)	112.3 ± 20.7a	107.7 ± 10.5a	97.8 ± 7.2a
BGB (g m ⁻²)	454.2 ± 52.5a	448.7 ± 38.2a	414.0 ± 31.1a
SOC (g kg ⁻¹ soil)	195.4 ± 6.7a	181.1 ± 2.8a	177.0 ± 5.7a
Soil moisture (wt %)	207.8 ± 11.2a	200.4 ± 11.2a	192.6 ± 3.8a
pH	5.7 ± 0.11a	5.7 ± 0.09a	5.8 ± 0.09a
Bulk density (g cm ⁻³)	0.29 ± 0.008a	0.29 ± 0.009a	0.30 ± 0.01a
Clay + silt (%)	63.2 ± 1.6a	65.5 ± 2.0a	67.2 ± 1.8a
Fe _d (g kg ⁻¹ soil)	17.9 ± 0.26a	16.0 ± 1.94a	15.2 ± 1.18a
Fe _o (g kg ⁻¹ soil)	11.8 ± 0.35a	10.1 ± 1.33a	10.4 ± 0.70a
Fe _p (g kg ⁻¹ soil)	4.0 ± 0.15a	3.8 ± 0.39a	3.9 ± 0.16a
OC-Fe (g kg ⁻¹ soil)	12.2 ± 0.91a	12.6 ± 0.56a	12.7 ± 0.53a
Illite (%)	29.6 ± 1.96a	30.6 ± 1.21a	29.2 ± 0.97a
I/S (%)	59.2 ± 1.50a	58.0 ± 1.45a	60.4 ± 1.21a
SSA (g m ⁻²)	9.7 ± 0.52a	10.0 ± 0.17a	9.3 ± 0.31a

The corresponding values are shown as means ± standard errors. Control #1, Control

#2 and Control #3 represent three non-collapsed plots which were paired to collapsed plots occurring for 1 year, 10 years and 16 years, respectively (for detailed plot distribution see Fig. R1). The moisture here is gravimetric soil moisture. AGB, aboveground biomass; BGB, belowground biomass; SOC, soil organic carbon; Clay + silt, the percentage of clay and silt; Fe_d, pedogenic Fe oxides; Fe_o, poorly crystalline Fe oxides; Fe_p, organically complexed Fe oxides; OC-Fe, Fe-bound organic carbon; I/S, mixed-layer illite/smectite; SSA, specific surface area in mineral-associated organic matter. Same letters represent no significant differences among the different plots (LSD test, $P > 0.05$).

Table R2. Characteristics of the sampling sites across the Tibetan Plateau.

Site	Coordinates	Elevation (m)	MAT (°C)	MAP (mm)	ALT (m)	Vegetation type	Soil type	Parent material	Mineralogy of clay (< 2 μm)	Thermokarst type
FEB	38.03°N, 100.89°E	3515	2.6	367	0.83	Swamp meadow	Cryosols	SS	Ilt, I/S, Chl, Kao	Thermo-erosion gully
SEB	38.00°N, 100.91°E	3650	2.6	367	0.71	Swamp meadow	Cryosols	SS	Ilt, I/S, Chl, Kao	Thermo-erosion gully
ML	37.76°N, 100.80°E	3760	1.4	411	1.10	Swamp meadow	Cryosols	SS	Ilt, I/S, Chl, Kao	Thermo-erosion gully
SLH	37.46°N, 100.28°E	3847	0.1	402	0.86	Swamp meadow	Cryosols	US	Ilt, I/S, Chl, Kao	Thermo-erosion gully
HSX	35.06°N, 98.71°E	4460	-1.0	353	0.82	Swamp meadow	Cryosols	US	Ilt, I/S, Chl, Kao	Thermo-erosion gully
HH	34.38°N, 97.95°E	4707	-3.1	436	0.78	Swamp meadow	Cryosols	SS	Ilt, I/S, Chl, Kao	Thermo-erosion gully

FEB, the first site at Ebo; SEB, the second site at Ebo; ML, site at Mole; SLH, site at Shaliuhe; HSX, site at Huashixia; HH, site at Huanghe; MAT, mean annual temperature; MAP, mean annual precipitation; ALT, active layer thickness. SS, siliciclastic sedimentary; US, unconsolidated sediments; Ilt, illite; I/S, mixed-layer illite/smectite; Chl, chlorite; Kao, kaolinite. The MAT and MAP data were derived from the National Meteorological Information Center (<http://data.cma.cn>). Soil type was classified based on the World Reference Base for Soil Resources (IUSS Working Group WRB, 2014). Parent material was identified according to the new global lithological map database GLiM (Hartmann and Moosdorf, 2012).

Table R3. Mean value and coefficient of variation of parameters from control plots (non-collapsed areas; for detailed plot information see Fig. R1) across the five thermokarst-impacted sites over the regional scale.

Site	SOC (g kg ⁻¹ soil)		Soil moisture (wt %)		pH		Clay+Silt (%)		Fe _d (g kg ⁻¹ soil)		Fe _o (g kg ⁻¹ soil)		Fe _p (g kg ⁻¹ soil)		Illite (%)		I/S (%)	
	Mean	CV(%)	Mean	CV(%)	Mean	CV(%)	Mean	CV(%)	Mean	CV(%)	Mean	CV(%)	Mean	CV(%)	Mean	CV(%)	Mean	CV(%)
FEB	98.3	6.1	102.3	11.1	6.6	1.7	62.4	2.7	15.3	8.0	9.6	6.8	2.9	13.1	28.2	11.0	64.2	5.6
SEB	129.5	11.2	153.0	14.4	6.6	4.6	68.3	4.1	13.2	10.6	8.5	13.2	1.9	11.6	41.8	8.5	47.8	6.8
ML	153.0	12.3	131.3	11.3	5.7	3.7	72.6	3.0	19.1	3.7	11.0	5.7	3.1	3.3	28.0	13.6	62.4	8.3
HSX	107.3	4.8	134.7	14.1	5.8	3.6	45.2	2.5	10.3	6.8	5.3	9.1	1.7	13.2	41.4	7.4	43.2	8.7
HH	90.6	7.6	139.6	7.3	6.4	5.4	42.3	7.9	12.3	14.2	6.2	13.1	2.2	11.2	34.0	5.5	54.2	5.1

FEB, the first site at Ebo; SEB, the second site at Ebo; ML, site at Mole; HSX, site at Huashixia; HH, site at Huanghe; CV, coefficient of variation (%); SOC, soil organic carbon; Clay+silt, the percentage of clay and silt; Fe_d, pedogenic Fe oxides; Fe_o, poorly crystalline Fe oxides; Fe_p, organically complexed Fe oxides; I/S, mixed-layer illite/smectite.

Table R4. Comparisons of biotic and abiotic properties between the non-collapsed (control) plot and the collapsed plot occurring for 1 year (the early stage of the permafrost thaw sequence; for detailed plot distribution see Fig. R1).

Variables	Control #1	Collapse _{1-year}
AGB (g m ⁻²)	112.3 ± 20.7a	108.2 ± 9.4a
BGB (g m ⁻²)	454.2 ± 52.5a	469.3 ± 28.8a
SOC (g kg ⁻¹ soil)	195.4 ± 6.7a	182.7 ± 3.7a
TN (g kg ⁻¹ soil)	16.2 ± 0.47a	15.9 ± 0.34a
pH	5.7 ± 0.11a	5.7 ± 0.06a
Bulk density (g cm ⁻³)	0.29 ± 0.008a	0.30 ± 0.02a
Clay + silt (%)	63.2 ± 1.6a	63.3 ± 1.4 a
Fe _d (g kg ⁻¹ soil)	17.9 ± 0.26a	16.1 ± 0.68 a
Fe _o (g kg ⁻¹ soil)	11.8 ± 0.35a	12.0 ± 0.29 a
Fe _p (g kg ⁻¹ soil)	4.0 ± 0.15a	4.0 ± 0.13 a
OC-Fe (g kg ⁻¹ soil)	12.2 ± 0.91a	11.8 ± 0.43 a
Illite (%)	29.6 ± 1.96a	30.0 ± 0.98a
I/S (%)	59.2 ± 1.50a	58.3 ± 1.14a
SSA (g m ⁻²)	9.7 ± 0.52a	9.7 ± 0.27a

The corresponding values are shown as means ± standard errors. Control #1 and Collapse_{1-year} denote the paired non-collapsed control and 1-year collapsed plots, respectively (for detailed plot distribution see Fig. R1). AGB, aboveground biomass; BGB, belowground biomass; SOC, soil organic carbon; TN, total nitrogen; Clay + silt, the percentage of clay and silt; Fe_d, pedogenic Fe oxides; Fe_o, poorly crystalline Fe oxides; Fe_p, organically complexed Fe oxides; OC-Fe, Fe-bound organic carbon; I/S, mixed-layer illite/smectite; SSA, specific surface area in mineral-associated organic matter. Same letters represent no significant differences between the two plots (LSD test, $P > 0.05$).

Step 1: Sampling along the thaw sequence

Step 2: Sampling across the regional sites

Fig. R1. Schematic diagram of soil sampling along the thaw sequence (SLH) and across the regional sites (marked as FEB, SEB, ML, HSX and HH) on the Tibetan Plateau. a, Location of the permafrost thaw sequence based on the map of permafrost distribution on the Tibetan Plateau (Zou et al., 2017). b, Image of the thermo-erosion gully captured from coloured LiDAR point cloud data (VZ-400, Riegl, Horn, Austria) with the specific plot distribution along the thaw sequence. Control #1, Control #2 and Control #3 represent three non-collapsed plots which were paired to collapsed plots occurring for 1 year, 10 years and 16 years, respectively. c, Sampling schematic diagram for the thaw sequence (photo credit: F.T. Liu). d, Distribution of five thermokarst-impacted sites at the regional scale. e, Landscapes of the regional sites with non-collapsed control and collapsed plots (photo credit: Z.L. Li). f, Sampling schematic diagram for the regional sites. SLH, site at Shaliuhe; FEB, the first site at Ebo; SEB, the second site at Ebo; ML, site at Mole; HSX, site at Huashixia; HH, site at Huanghe. Notably, along the thaw sequence, the distance between the collapsed plots occurring for 1 year and those for 10 and 16 years is 80 m and 130 m respectively, while the distance between the collapsed and non-collapsed control plots is less than 1 m. Across the five sites over the regional scale, we set up one paired collapsed and control plots (15 × 10 m) at the end of a gully and in adjacent non-collapsed areas at each site. These five sites were distributed across a 550-km permafrost transect on the Tibetan Plateau. Under our sampling design, there were five replicates in the control plot and ten replicates in the collapsed plots along the thaw sequence. Across the five thermokarst-impact sites over the regional scale, there were five replicates in both control and collapsed plots.

Fig. R2. Changes in the absolute contents of POC and MAOC induced by thermokarst formation. a-b, Variations in contents of POC (a) and MAOC (b) along the thaw sequence. c-d, Comparisons of POC (c) and MOAC (d) contents between collapsed (dark colour) and control (light colour) plots at each of the five regional thermokarst-impacted sites: FEB, SEB, ML, HSX and HH. POC, particulate organic carbon; MAOC, mineral-associated organic carbon. FEB, the first site at Ebo, SEB, the second site located at Ebo. ML, HSX and HH indicate the sites at the Mole, Huashixia and Huanghe, respectively. Different capital letters indicate significant differences for the variables within plots along the thaw sequence (LSD test, $P < 0.05$). Dashed lines distinguish different thermokarst-impacted sites, denoting that the parameters are compared between collapsed and control plots in each site rather than across various study sites. * $P < 0.05$ and ** $P < 0.01$.

Fig. R3. Picture showing vegetated patches (marked with white border) and exposed patches (marked with yellow border) within the collapsed plot across the Tibetan thermokarst-impacted sites (photo credit: F.T. Liu). The inset shows topsoil sampling in vegetated patches. Notably, to avoid the interference of soil layer mixture, we collected topsoil (0-15 cm) within the vegetated patches (40-60 cm thickness) rather than within the exposed patches, in which soil cores were at least 10 cm away from the edge of vegetated patch.

Fig. R4. Picture showing matic epipedon in swamp meadow on the Tibetan Plateau (photo credit: F.T. Liu). The depth of matic epipedon ranges from 40 to 60 cm across the thermokarst-impacted sites.

Minor comments:

[Comment 3] line 76-80 Why would be an enhanced mineral association of OC be bad in this respect? If climatic stabilization vanishes, MAOM might be one key pool for persistent OC.

[Response] Sorry for the inappropriate description in the original MS. To avoid this confusion, **we have revised this sentence as follows**: “*Consequently, soil C emissions could be reduced if the amount of MAOC or OC-Fe increases under permafrost thaw. Accordingly, changes in these mineral-associated organic C fractions would strengthen the stability of the entire soil C pool and weaken the potential permafrost*”

C-climate feedback” (Page 4, line 76-80).

[Comment 4] line 85 thermokarst comes in a lot of forms, thus it is too reductionist in this respect, please specify.

[Response] We agree with the reviewer that thermokarst comes in a lot of forms and our thermokarst sites involved in this study is only one form of them. Hence, **we have re-organized this sentence as follows:** “*POC content may exhibit substantial decline due to the improved soil aeration and microbial C processing caused **by abrupt permafrost thaw occurring in uplands like thermo-erosion gullies** (Abbott et al., 2015; Schuur et al., 2015) (Page 5, line 84-86).*

[Comment 5] line 87-88 It is more the other way around, OM sorbes to mineral surfaces.

[Response] Following the reviewers’ comment, **we have rephrased this sentence as follows:** “*MAOC content, which is difficult for decomposers to degrade due to **the adsorption of organic C to soil minerals** (Lavallee et al., 2020; Patzner et al., 2020), would remain stable upon permafrost thaw” (Page 5, line 87-88).*

[Comment 6] line 94-97 How comparable are these sites in terms of soil type, parent material, elevation, vegetation etc.? Provide data to support this assumption.

[Response] We would like to mention that the space for time approach was **only applied for the permafrost thaw sequence located at Shaliuhe**, which was not used for the other five sites located at Ebo, Mole, Huashixia and Huanghe. Along the thaw sequence, we analysed a series of parameters (*i.e.*, aboveground biomass, belowground biomass, SOC, soil moisture, pH, bulk density, soil texture and soil minerals) among the three control plots which were located outside the gully but adjacent to three collapsed plots within the gully (Table R1). **We observed that the above parameters were comparable, illustrating that the permafrost thaw gradient met the requirements of space for time approach.** We have clearly stated these points in the revised MS (Page 18, line 370-388).

Across these five sites over the regional scale, we mainly focused on the impact of permafrost collapse on POC and MAOC inside and outside the gully in each site, without involving the space for time method and the comparisons among these five sites. Given this point, the pristine attributes between sites, such as vegetation, soil type, parent material, elevation, clay minerals and soil texture, do not have to be similar and comparable. Nevertheless, **following the reviewer’s suggestion, we have provided this information for our sites in the** Supplementary Figure 1 and Supplementary Table 2 of the revised MS (Pages 2-3, line 5-16; Page 16, line 92-97 in the supplementary materials). Based on these data, we found that the pristine soils in each site could also be regarded as homogeneous because of the low coefficient of variation of parameters in the control plot of each site (Table R3; Wiltshire et al., 1986). These selected sites were basically in line with the experimental purpose of this study since the parameters between collapsed and control plots were comparable, and the differences in parameters inside and outside gully could be attributed to the effects of permafrost collapse. We have clearly stated this point in the revised MS (Page 18, line 370-378).

[Comment 7] line 134 - 136 Please give explanation for the site names / at least name them in the figure caption. In the text itself it always hampers the reading flow having such abbreviations, thus try to avoid them.

[Response] Following the reviewers’ suggestion, we have given the explanation for the site names in the figure caption of the revised MS as follows: “*FEB, the first site at Ebo, SEB, the second site located at Ebo. ML, SLH, HSX and HH indicate the sites at Mole, Shaliuhe, Huashixia and Huanghe, respectively*” (Page 37, line 794-796 and line 806-807; Page 39, line 840-842). Moreover, we also provided full name rather than abbreviation of these study sites in the main text of revised MS to improve its readability (Pages 6-9, line 125-174).

[Comment 8] line 138 Please give information on soil texture and mineralogy (clay

minerals) of the studied soils to demonstrate comparability.

[Response] As mentioned above, the approach involving space for time was mainly used for the permafrost thaw sequence, rather than the other five sites over the regional scale. Hence, we need to prove that the three control plots (adjacent to three collapsed plots occurring for 1 year, 10 years and 16 years) along the thaw sequence are comparable. Given this point, we examined a series of soil parameters including soil texture and clay minerals, and confirmed that three control plots were similar (Table R1; Pages 10-11, line 67-76 in the supplementary materials), and thus the collapsed plots induced by thermokarst were comparable. We have clearly stated these points in the revised MS (Page 18, line 370-388).

Nevertheless, we have also added the data involving vegetation, soil type, parent material, elevation, soil texture and clay minerals from these six sites in the Supplementary Figure 1 and Supplementary Table 2 of the revised MS (Pages 2-3, line 5-16; Page 16, line 92-97 in the supplementary materials). Based on these data, we found that the pristine soils in each site could also be regarded as homogeneous because of the low coefficient of variation of parameters in the control plot of each site (Table R3; Wiltshire et al., 1986). These selected sites were basically in line with the experimental purpose of this study since the parameters between collapsed and control plots were comparable, and the differences in parameters inside and outside the gully could be attributed to the effects of permafrost collapse. Given that this study mainly focused on the impact of permafrost collapse on POC and MAOC by comparing soil C fractions inside and outside the gully in each site rather than among the study sites, the differences in elevation and soil texture between these sites would not affect the reliability of our results. We have clearly stated these points in the revised MS (Page 18, line 370-378). Thanks for your understanding!

[Comment 9] line 142-144 Thus, as there were clear differences in the texture between the soils, how do POM / MAOM ratios relate between them? Texture is a clear driver

for the fate of POM vs. MAOM, thus it has to be discussed how textural differences interfere with the overall shifts, and or control contents.

[Response] As mentioned above, the space for time approach was **only applied for the permafrost thaw sequence (Shaliuhe site)**. In this site, **we observed that soil texture was not significantly different among the three control plots** which were located outside the gully but adjacent to three collapsed plots within the gully (Table R1), **demonstrating that soil texture was comparable and the permafrost thaw sequence met the requirements of space for time approach.** We have clearly stated this point in the revised MS (Page 18, line 382-386).

Again, we would like to mention that **we did not use the space for time approach for the other five sites at the regional scale.** Across these five sites, we mainly focused on the impact of permafrost collapse on POC and MAOC inside and outside the gully in each site, **rather than comparing SOC fractions among various sites.** Given this point, the differences in soil texture among these five sites will not interference our result interpretation. We have clearly stated this point in the revised MS (Page 18, line 370-378).

[Comment 10] line 153 *OM-Fe assemblages are only one means of forming MAOM, thus it only represents a part of the whole MAOM pool.* line 154-155 *Mainly indicating that in drier soils the amount of FeIII oxides is higher and thus there is more OC association with more stable forms of FeO (see Patzner et al. Nat Comms. 2020).*

[Response] We do agree with the reviewer's opinion that OM-Fe assemblages are only one means of forming MAOM. Given this point, **we have revised this sentence in the revised MS as follows:** "*The organically complexed Fe oxides were positively correlated with OC-Fe ($P < 0.001$; Fig. 4e), indicating that the increased Fe oxides and improved soil aeration induced by permafrost collapse could result in the accumulation of OC-Fe along the thaw sequence*" (Page 9, line 187-190). In addition, we have discussed this issue in the *Discussion* section of the revised MS (Pages 13-14,

line 260-291).

[Comment 11] line 163-165 Soil enzymes are more or less an indicator of stoichiometric needs of microbiota, so it's questionable if this is really causation. Whereas, soil moisture directly links with microbial activity and thus the bioavailability of OM, especially POM.

[Response] We agree with the reviewer that soil enzymes cannot be recognized as a factor influencing the variations of POC, while soil moisture directly links with microbial activity and degradation for POC. Given this point, **we deleted enzyme data and the associated discussions about the effects of enzyme activity on POC, and highlighted the critical role of soil moisture in affecting POC** in the *Discussion* section of the revised MS (Pages 11-12, line 232-238).

[Comment 12] line 169-170 These are not "regulating" MAOC. OC-Fe indicates just that Fe is important for OM storage in the MAOM fraction, it is not "regulating" it. And microbial necromass is one major contributor to MAOM (see work on microbial residues as precursors for MAOM). Thus, if MAOM is high, also microbial necromass should be high and vice versa.

[Response] We agree with the reviewer's opinion that OC-Fe cannot be regarded as regulatory factors for MAOM. In the revised MS, we used microbial necromass and soil minerals suggested by the reviewer, rather than OC-Fe, to explain the dynamics of MAOC induced by thermokarst formation (Pages 8-9, line 165-174; Page 12, line 238-244; Page 12, line 251-258).

[Comment 13] line 192-195 How comparable are the soil layers / horizons that were studied? Did thaw processes lead to an increased admixing of top and subsoils and thus a dilution of POM?

[Response] We clarified that **soil samples were collected from 0-15 cm soil layer and soil physical mixing did not occur in the vegetation patches within collapsed plots,**

soil layers between the collapsed and control plots were comparable. To be specific, as done in previous studies (Abbott and Jones, 2015; Mu et al., 2016; Yuan et al., 2018), we only selected topsoil (0-15 cm) samples within the vegetated patches (40-60 cm thickness) rather than from exposed soil areas in the collapsed sites (Fig. R3). Given that these collected soil samples in this swamp meadow are protected by matic epipedon (which has an intensive root network protecting soils against interference; Fig. R4) (Jiang et al., 2020; Ma et al., 2020), they are not easy to be mixed with the subsoil.

To confirm this point, we compared the non-collapsed (control) plot with the collapsed plot occurring for 1 year (the early stage of the permafrost thaw sequence), and observed no significant differences in soil properties such as **bulk density, SOC, pH, soil texture and soil minerals** (all $P > 0.05$; Table R4). These comparisons illustrated that permafrost collapse did not cause soil physical mixing for topsoil samples, and thus **the variations of POC could be ascribed to the effect of permafrost collapse rather than the dilution effect**. We have clearly stated this point in the revised MS (Pages 18-19, line 390-408).

[Comment 14] line 212-217 POC is not necessarily more accessible than MAOC. There are for instance occluded POM fractions that are known to be rather inaccessible to microbiota due to their spatial arrangement in the soil. However, the less decomposed POM is, which means the higher the plant derived OM content, the better bioavailable is the POM. Thus, it is basically the chemical composition of the POM that drives its decomposition. An indication for that in your data is the C/N ratio.

[Response] We agree with the reviewer's opinion that POC is not necessarily more accessible than MAOC, and thus have deleted these unreasonable expressions in the revised MS.

[Comment 15] line 219-221 Although enzymes can be long lived, these values present a short time information about the microbial OC consumption. The POC to MAOC

ratio is the result of longer lasting decomposition processes. Thus, enzyme activity is more an indicator of the current environmental (e.g. aerobic vs. anaerobic conditions) and nutrient availability (stoichiometry) conditions in the respective soils. line 227-229 These environmental factors triggered specific shifts in microbial OM consumption which lead to differences in enzyme production. Enzymes per se are not active entities that react to for instance altered accessibility of OM or stoichiometric needs of microbiota.

[Response] We agree with the reviewer that enzyme activity is an indicator of the current environmental and nutrient availability conditions which could not be used to explain the variations of POC because of longer lasting decomposition processes. Therefore, **we removed this parameter and associated descriptions in the revised MS, and highlighted the important role of thermokarst-induced shifts in environmental factors in influencing POC** (Pages 11-12, line 230-238).

[Comment 16] line 232 Although it was reported before, assumed from the recovery of occluded POM, aggregation per se might not be a key process in these soils at initial stages. Aggregation might play a larger role in more degraded mineral rich soils after permafrost thaw.

[Response] We agree with the reviewer's opinion and have removed this sentence in the revised MS.

[Comment 17] line 241-243 The main driver for the increased decomposition of especially POM rich OC pools is, the higher aeration, and higher temperatures. One can assume that the POM is not yet more stabilized due to aggregation in these permafrost affected soils.

[Response] Following the reviewer's suggestion, we have revised this sentence as follows: "Consistent with this deduction, positive associations were observed between POC content and soil moisture along the thaw sequence and also between thaw-induced shifts in POC content and the corresponding changes in soil moisture across the

regional thermokarst-impacted sites (Fig. 4b-c), reflecting that **the decreased soil moisture and improved soil aeration induced by permafrost collapse could stimulate microbial decomposition and thus lead to the POC loss**" (Page 11, line 232-238).

[Comment 18] line 245-248 Please clearly show that there is an actual increase in MAOC and not just a relative increase due to decreasing amounts of POC (see data in supplement). As the MAOC C contents are decreasing with permafrost collapse it seems there is a deepening of the profiles that leads to an admixing with minerals that are lower in OC.

[Response] **Regarding the first issue, as mentioned above, we examined the absolute content of MAOC and found that there was no significant difference in MAOC content** between the non-collapsed and collapsed soils across most of the thermokarst-impacted sites. Specifically, we found that there was no significant difference in MAOC content along the thaw sequence or in four of the five sites at the regional scale. Only one site, located at Mole, exhibited a decreased C content of MAOC with collapse time, which may be ascribed to MAOC degradation (Hall et al., 2015). Given this point, we agree with the reviewer that the increase of proportion of MAOC to SOC may be attributed to the decline of POC. Therefore, **in the revised MS, we not only focused on the proportion of POC and MAOC, but also emphasized the divergent responses of POC and MOAC contents to permafrost thaw**. We have added the data involving the absolute contents of POC and MAOC, and modified the descriptions of *Results* and *Discussion* sections in the revised MS (Page 6, line 109-123; Pages 11-12, line 226-258). Thanks for your understanding!

Regarding the second issue, we would like to mention that our sampling method could avoid the problem mixing of upper and lower soil layers. As mentioned above in our response to [Comments 2 and 13], we only collected topsoil (0-15 cm) samples within the vegetated patches (40-60 cm thickness) rather than the exposed soils in the collapsed sites. Given that these collected soil samples are protected by matic epipedon

in this swamp meadow ecosystem (Fig. R4; Jiang et al., 2020; Ma et al., 2020), they would not lead to an admixing with minerals that are lower in OC. To further illustrate this point, we also compared the non-collapsed (control) plot with the collapsed plot occurring for 1 year in the permafrost thaw sequence, and found that there was no significant differences in soil properties such as bulk density, SOC, pH, soil texture and soil minerals (Table R4), demonstrating that this thaw processes would not lead to an admixing of top and subsoils for those vegetated patches. We have clearly stated this point in the revised MS (Page 18-19, line 390-408).

[Comment 19] line 270 by Fe/Al oxides, but especially also by clay minerals!

[Response] Following the reviewer's comment, we have revised this sentence as follows: "MAOC is protected by Fe oxides and clay minerals through sorptive interactions" (Page 12, line 238-239). In addition, we have determined clay minerals and discussed their critical roles in affecting MAOC dynamics upon permafrost thaw in the revised MS (Page 8, line 165-171; Pages 11-12, line 238-244).

[Comment 20] line 270-272 This sentence is hard to understand, please rephrase.

[Response] Sorry for the poor description. Following the reviewer's comment, we have rephrased this sentence as follows: "In particular, clay-sized particles including Fe oxides and phyllosilicates can provide a high specific surface area, and thus adsorb substantial amounts of SOC (von Lützow et al., 2006; Wiesmeier et al., 2019; Chen et al., 2020)" (Page 12, line 239-241).

[Comment 21] line 273-274 What has this to do with your data? Yes, mineral surfaces have charged surfaces that can be act as sites for OM sorption. But as you have no information about specific surface area or other mineralogical data this is just speculative.

[Response] Following the reviewer's comment, we have determined the specific surface area and other mineralogical data such as clay minerals. Our results

revealed that clay minerals mainly consisted of illite, mixed-layer illite/smectite, chlorite and kaolinite in our study sites (Supplementary Table 1), and the former two accounted for more than 80% of clay mineral composition (Supplementary Fig. 1j-k). However, both the illite and mixed-layer illite/smectite minerals showed no significant changes along the thaw sequence or at the five thermokarst-impacted sites over the regional scale. Furthermore, the specific surface area of soil mineral-associated organic matter also showed no significant changes at most of the regional thermokarst-impacted sites. The variations of these parameters including clay minerals and SSA are logically consistent with relatively stable MAOC upon permafrost thaw. We have added this data and related descriptions in the revised MS (Pages 8-9, line 165-174; Page 12, line 238-244).

[Comment 22] line 274-276 This is mixing up different concepts. Plant roots and hyphae especially foster macroaggregation (see literature for root/fungal effects on macroaggregation), whereas MAOM is enriched in microaggregates.

[Response] We agree with the reviewer that plant roots and hyphae especially foster macroaggregation while MAOM is enriched in microaggregates (Lavalley et al., 2020; Cotrufo et al., 2022). Microaggregates were primarily related with the Fe/Al oxides and clay minerals (Lavalley et al., 2020; Cotrufo et al., 2022). **Therefore, we have removed this description in the revised MS.**

[Comment 23] line 276-279 Again, is this a total increase or just due to relative effects of lower POM and changes in admixing with different soil material?

[Response] As mentioned above, our results revealed that the absolute content of MAOC exhibited no significant difference before and after permafrost collapse, and thus the increased proportion of MAOC may be due to the decrease of POC. We have revised the related descriptions in the *Results* and *Discussion* sections in the revised MS, **in which we not only focused on the proportion of POC and MAOC, but also analysed the dynamics of POC and MOAC contents upon permafrost thaw** (Pages

6-7, line 109-133; Pages 11-12, line 226-258; Pages 14-15, line 293-307). Thanks for your understanding!

Moreover, as mentioned above, given that our soil samples were collected from surface soils (0-15 cm) protected by matic epipedon which has an intensive root network protecting soils against interference (Fig. R4; Jiang et al., 2020; Ma et al., 2020), the variations of MAOC would be less affected by mixing with subsoils, but largely influenced by permafrost collapse. To confirm this point, we compared the non-collapsed control plot with the collapsed plot occurring for 1 year, and **found that there was no significantly differences in soil properties such as bulk density, SOC, pH, soil texture and soil minerals** (Table R4), illustrating that **permafrost thaw would not lead to an admixing of top and subsoils in our case**. We have clearly stated this point in the revised MS (Pages 18-19, line 390-408).

[Comment 24] line 358-361 With such an active layer depth I would assume that the soils rather classify as Cryosols given the massive permafrost layer within the profile depth.

[Response] Following the reviewer's comments, we have carefully read the World Reference Base for Soil Resources and figured out the definition between Cambisols and Cryosols. According the definition, Cambisols are characterized by slight or moderate weathering of parent material and by absence of appreciable quantities of illuviated clay, organic matter, Fe and/or Al compounds. **Cryosols have a cryic horizon starting \leq 100 cm from the soil surface or a cryic horizon starting \leq 200 cm from the soil surface and evidence of cryoturbation (frost heave, cryogenic sorting, thermal cracking, ice segregation, patterned ground, etc.) in some layer within \leq 100 cm of the soil surface.** Given the shallower active layer thickness (\leq 100 cm), higher content of Fe oxides and occurrence of cryoturbation (*i.e.*, thermokarst) in less than 100 cm soil depth in our study sites, we agree with the reviewer that the soils should be classified as Cryosols rather than Cambisols. Therefore, we have revised this

sentence as follows: “The main soil type is Cryosols according to the World Reference Base for Soil Resources (IUSS Working Group WRB, 2014)” (Pages 15-16, line 325-326).

[Comment 25] line 370-371 How were the filtered particles prepared for C analysis. Was also N determined? If so, please report C/N ratios for all OM fractions (see also comments above).

[Response] The filtered particles was prepared as the following procedure. **First**, the filtered particles were completely washed with deionized water. **Second**, these washed particles were oven-dried to constant weight at 60 °C. **Third**, the dried particulate organic matter was fully grinded for C analysis. We have clearly described these points in the revised MS (Page 20, line 415-418).

We would like to mention that **both C and N concentration in POM and MAOM were determined using an elemental analyzer** (Vario EL III, Elementar, Hanau, Germany) in this study. Then we calculated the carbon: nitrogen ratio of particulate and mineral-associated organic matter, respectively. Specifically, the carbon: nitrogen ratios of particulate organic matter significantly decreased along the thaw sequence and at four of five sites over the regional scale (all $P < 0.05$; Table R5). By comparison, the carbon: nitrogen ratio in mineral-associated organic matter significantly increased along the thaw sequence and at the two sites at Ebo town ($P < 0.05$) while that in another three sites showed no changes after permafrost collapse ($P > 0.05$; Table R5). **We have provided the data of C/N ratios for all OM fractions in the Supplementary Table 1, and also added the related descriptions in the revised MS (Page 7, line 133-139).**

Table R5. Mass distribution, carbon concentration and C/N ratios in particulate organic matter (POM), heavy particulate organic matter (HPOM) and mineral-associated organic matter (MAOM) across the thermokarst-impacted sites on the Tibetan Plateau.

Site	Plot	Weight of POM (%)	POM carbon concentration (g kg ⁻¹ POM)	POM C/N ratio	Weight of HPOM (%)	HPOM carbon concentration (g kg ⁻¹ HPOM)	HPOM C/N ratio	Weight of MAOM (%)	MAOM carbon concentration (g kg ⁻¹ MAOM)	MAOM C/N ratio
SLH	Control	3.4 ± 0.4a	346.3 ± 13.6a	19.8 ± 1.7a	34.6 ± 0.8a	236.2 ± 2.5a	12.5 ± 0.1a	51.8 ± 1.6c	148.6 ± 3.9a	10.8 ± 0.2b
	1 year	3.5 ± 0.2a	381.4 ± 9.2a	20.0 ± 0.8a	31.2 ± 1.3b	246.9 ± 4.5a	12.5 ± 0.1a	59.8 ± 1.2b	137.1 ± 4.3b	10.8 ± 0.2b
	10 years	2.8 ± 0.2ab	263.2 ± 17.0b	18.8 ± 1.4ab	30.1 ± 1.4b	233.3 ± 2.8a	12.4 ± 0.1a	61.6 ± 1.6b	129.8 ± 2.9b	10.9 ± 0.1b
	16 years	2.2 ± 0.3b	255.1 ± 12.5b	18.4 ± 0.9b	25.3 ± 1.4c	207.2 ± 15.0b	12.1 ± 0.2a	69.2 ± 1.5a	104.8 ± 4.7c	11.2 ± 0.2a
FEB	Control	1.1 ± 0.1a	201.1 ± 10.9a	22.3 ± 1.1a	20.5 ± 3.2a	171.9 ± 16.1a	13.8 ± 0.2a	67.3 ± 3.8a	89.4 ± 6.1a	11.2 ± 0.1b
	Collapse	0.8 ± 0.1b	170.1 ± 5.8b	21.0 ± 0.5b	14.9 ± 1.1a	118.5 ± 6.6b	13.2 ± 0.3a	76.6 ± 2.1a	86.3 ± 6.4a	11.6 ± 0.2a
SEB	Control	3.5 ± 0.5a	270.8 ± 12.0a	21.8 ± 1.8a	34.8 ± 1.2a	168.3 ± 7.5a	13.3 ± 0.1a	54.7 ± 2.4a	103.1 ± 12.2a	10.9 ± 0.2b
	Collapse	2.1 ± 0.3b	191.5 ± 10.1b	19.6 ± 1.0b	31.3 ± 1.5a	108.0 ± 2.3b	12.8 ± 0.2a	56.4 ± 2.0a	95.0 ± 5.5a	11.3 ± 0.1a
ML	Control	3.8 ± 0.4a	289.3 ± 11.0a	24.3 ± 0.6a	28.2 ± 2.6a	225.5 ± 11.2a	13.8 ± 0.3a	61.1 ± 2.8b	126.7 ± 5.0a	11.3 ± 0.3a
	Collapse	1.8 ± 0.4b	236.1 ± 18.1b	22.6 ± 1.4b	23.5 ± 1.6a	167.6 ± 5.9b	13.2 ± 0.2a	70.5 ± 1.2a	95.1 ± 1.0b	11.5 ± 0.3a
HSX	Control	1.2 ± 0.2a	259.4 ± 15.1a	20.0 ± 1.5a	39.2 ± 1.1a	156.6 ± 5.4a	12.9 ± 0.1a	46.8 ± 1.3b	84.2 ± 3.5a	10.3 ± 0.2a
	Collapse	0.8 ± 0.1b	165.2 ± 11.9b	19.1 ± 1.9a	28.9 ± 2.0b	104.7 ± 4.9b	12.3 ± 0.4a	60.5 ± 2.2a	64.7 ± 3.2b	10.1 ± 0.3a
HH	Control	3.1 ± 0.7a	198.3 ± 17.7a	21.1 ± 1.6a	35.7 ± 0.9a	102.6 ± 2.2a	13.4 ± 0.2a	49.1 ± 1.3a	97.8 ± 4.7a	10.9 ± 0.3a
	Collapse	0.9 ± 0.3b	182.3 ± 5.2a	19.2 ± 0.4b	38.7 ± 2.0a	92.1 ± 11.0a	13.4 ± 0.2a	48.1 ± 1.6a	85.5 ± 8.0a	10.6 ± 0.2a

Data are means ± standard errors. SLH, site at Shaliuhe; FEB, the first site at Ebo; SEB, the second site at Ebo; ML, site at Mole; HSX, site at Huashixia; HH, site at Huanghe. The carbon concentration of POM, HPOM and MAOM means the quantities of carbon normalized to unit POM, HPOM, and MAOM. C/N ratio represent the ratio of organic carbon to total nitrogen. Different letters indicate significant differences between collapse and control plots (LSD test, $P < 0.05$).

[Comment 26] line 390-391 Please explain how OM and carbonates were removed. Please give texture data for all soils, not only the ones normalized by OC.

[Response] Following the reviewer's suggestion, **we have rephrased the related sentences to clarify how we removed OM and carbonates (Page 21, line 450-453) and supplied the texture data in Supplementary Figure 1 of the revised MS (Pages 2-3, line 5-16 in the supplementary materials).** To be specific, **we used hydrogen peroxide (30%) to remove OM, and utilized hydrochloric acid (3 M) to remove carbonates** (Chen et al., 2016; Igaz et al., 2020). The specific procedure has been shown as follows: "First, we added 100 mL 30% hydrogen peroxide to the breakers with soil sample, which was then heated at 80 °C for 4 h. After standing overnight, the solution was added with 100 mL deionized water and heated to boiling for 1 h to remove redundant hydrogen peroxide. Second, 10 mL hydrochloric acid (3 M) and 100 mL deionized water were added to the solution, which was then heated to boiling for 0.5 h to remove carbonates. Third, after adjusting pH to neutral, we added 10 ml sodium hexametaphosphate (0.05 M) to the solution with boiling for 5 min. After cooling to room temperature, the treated solution was examined using a particle size analyzer (Malvern Masterizer 2000, Malvern, Worcestershire, UK)".

[Comment 27] Supplement Table 2, The changes in the C content of the POM look like there was a change in admixing of minerals in the POM separates. When fractionating a clean POM the composition and thus the C content should stay approx comparable. It would be good to check the C/N ratios to better describe what lead to this change in C contents.

[Response] Regarding the reviewer's concern about the admixing of minerals in the POM separates, we would like to clarify this issue following the three aspects. **First, we separated POM and MAOM from soils strictly following the fractionation method (Lavallee et al., 2020),** which can clearly distinguish POM and MAOM and avoid the mixing of them. Specifically, POM and MAOM were separated based on the combination of density (1.6 g cm^{-3}) and particle size ($53 \text{ }\mu\text{m}$). At the first step, the

floating particulate organic matter could be separated from soils by using 1.6 g cm^{-3} NaI and GF/C filter membrane (Fig. R5). At the second step, the residual soils ($>1.6 \text{ g cm}^{-3}$) was further separated by using a $53\text{-}\mu\text{m}$ sieve, and only the organic matter through the sieve ($< 53 \mu\text{m}$) was collected as mineral-associated organic matter. Therefore, POM and MAOM are not easy to mix following the fractionation procedure.

Second, to confirm the reliability of our research data, **we compared the C concentration of POM in this study with that reported by other studies, and found that our data were within a reasonable range** (Table R6), which indirectly indicated that POM was not mixed with MAOM.

Third, following the reviewer's suggestion, we have analysed the C/N ratios in all collected soil samples. Our results revealed that permafrost collapse led to a significant decrease in C/N ratio of POM, whether along the thaw sequence or across four of the five thermokarst-impacted sites over the regional scale (Table R5). **Decreased C/N ratios of POM suggested that permafrost collapse stimulated POM decomposition with collapse time** (Vogel et al., 2015). Given that soil samples were protected by matic epipedon in this alpine swamp ecosystem on the Tibetan Plateau (which has an intensive root network shielding soils against interference; Fig. R4) (Jiang et al., 2020; Ma et al., 2020), the decreased C concentration of POM could be attributed to the degradation of POM (Angst et al., 2017) rather than the mixing of top- and subsoil. To eliminate the concern of reviewers and other readers, we have provided the C/N ratios of soil samples in the Supplementary Table 1 of revised MS (Pages 14-15, line 84-91 in the supplementary materials).

Fig. R5. Picture showing the separation between POM and MAOM based on the fractionation method (photo credit: F.T. Liu). POM, particulate organic matter; MAOM, mineral-associated organic matter.

Table R6. Comparisons of POM carbon concentration between published references and this study.

Vegetation type	POM carbon concentration (g kg ⁻¹ POM)	References
Swamp meadow	165.2-381.4	This study
Tundra	189.8-429.7	Prater et al., 2020
Grassland	214.5-224.9	Bai et al., 2020
Grassland	128.8-296.4	Wiesmeier et al., 2012
Forest	243.9-339.1	Lajtha et al., 2014
Forest	182.3-315.5	Angst et al., 2017

Thanks again for this reviewer’s insightful and professional review. These comments inspired us to have a deeper thinking on both field sampling, data analyses and results integration, and thus guided us to conduct a thorough revision of the original MS. To address these insightful comments, **we described the field sampling strategies in more detail, demonstrated the comparability of plots along the thaw sequence, soil horizons involved in our study were little affected by physical mixing and translocation, analysed the absolute changes in POC and MAOC content, and discussed the potential mechanisms for POC and MAOC in response to permafrost collapse. Particularly, we added another eight supplementary figures and three supplementary tables to support the main conclusion drawn in this study.** By doing so, we feel that our revised manuscript has been greatly improved and expect that the reviewer will be satisfied with the revised manuscript. Thank you!

References

- Abbott, B. W. & Jones, J. B. Permafrost collapse alters soil carbon stocks, respiration, CH₄, and N₂O in upland tundra. *Glob. Chang. Biol.* **21**, 4570-4587 (2015).
- Angst, G., Mueller, K. E., Kögel-Knabner, I., Freeman, K. H. & Mueller, C. W. Aggregation controls the stability of lignin and lipids in clay-sized particulate

- and mineral associated organic matter. *Biogeochemistry* **132**, 307-324 (2017).
- Bai, T. et al. Interactive global change factors mitigate soil aggregation and carbon change in a semi-arid grassland. *Glob. Chang. Biol.* **26**, 5320-5332 (2020).
- Chen, C., Hall, S. J., Coward, E. & Thompson, A. Iron-mediated organic matter decomposition in humid soils can counteract protection. *Nat. Commun.* **11**, 2255 (2020).
- Chen, L. et al. Determinants of carbon release from the active layer and permafrost deposits on the Tibetan Plateau. *Nat. Commun.* **7**, 13046 (2016).
- Cotrufo, M. F., Haddix, M. L., Kroeger, M. E. & Stewart, C. E. The role of plant input physical-chemical properties, and microbial and soil chemical diversity on the formation of particulate and mineral-associated organic matter. *Soil Biol. Biochem.* <https://doi.org/10.1016/j.soilbio.2022.108648> (2022).
- Cotrufo, M. F. & Lavellee, J. M. Soil organic matter formation, persistence, and functioning: A synthesis of current understanding to inform its conservation and regeneration. *Adv. Agr.* **172**, 1-66 (2022).
- Cotrufo, M. F. et al. Formation of soil organic matter via biochemical and physical pathways of litter mass loss. *Nat. Geosci.* **8**, 776-779 (2015).
- Cotrufo, M. F., Wallenstein, M. D., Boot, C. M., Deneff, K. & Paul, E. The microbial efficiency-matrix stabilization (MEMS) framework integrates plant litter decomposition with soil organic matter stabilization: Do labile plant inputs form stable soil organic matter? *Glob. Chang. Biol.* **19**, 988-995 (2013).
- Fulton-Smith, S. & Cotrufo, M. F. Pathways of soil organic matter formation from above and belowground inputs in a *Sorghum bicolor* bioenergy crop. *Glob. Chang. Biol. Bioen.* **11**, 971-987 (2019).
- Gentsch, N. et al. Properties and bioavailability of particulate and mineral-associated organic matter in Arctic permafrost soils, Lower Kolyma Region, Russia. *Eur. J. Soil Sci.* **66**, 722-734 (2015).
- Gentsch, N. et al. Temperature response of permafrost soil carbon is attenuated by mineral protection. *Glob. Chang. Biol.* **24**, 3401-3415 (2018).

- Hall, S. J., McNicol, G., Natake, T. & Silver, W. L. Large fluxes and rapid turnover of mineral-associated carbon across topographic gradients in a humid tropical forest: insights from paired ^{14}C analysis. *Biogeosciences* **12**, 2471-2487 (2015).
- Hartmann, J. & Moosdorf, N. The new global lithological map database GLiM: A representation of rock properties at the Earth surface. *Geochem. Geophys. Geosys.* **13**, 1-37 (2012).
- Hodgkins, S. B. et al. Changes in peat chemistry associated with permafrost thaw increase greenhouse gas production. *Proc. Natl Acad. Sci. USA* **111**, 5819-5824 (2014).
- Huang, W. & Hall, S. J. Elevated moisture stimulates carbon loss from mineral soils by releasing protected organic matter. *Nat. Commun.* **8**, 1774 (2017).
- Igaz, D., Aydin, E., Šinkovičová, M., Šimanský, V., Tall, A. & Horák, J. Laser diffraction as an innovative alternative to standard pipette method for determination of soil texture classes in central Europe. *Water* **12**, 1232 (2020).
- IUSS Working Group WRB: World reference base for soil resources 2014. International soil classification system for naming soils and creating legends for soil maps (FAO, Rome, 2014).
- Jiang, X., Zhu, X., Yuan, Z., Li, X. & Zakari, S. Lateral flow between bald and vegetation patches induces the degradation of alpine meadow in Qinghai-Tibetan Plateau. *Sci. Total Environ.* **751**, 142338 (2020).
- Joergensen, R. G. The fumigation-extraction method to estimate soil microbial biomass: Calibration of the k_{EN} value. *Soil Biol. Biochem.* **28**, 25-31 (1996).
- Joss, H., Patzner, M. S., Maisch, M., Mueller, C. W., Kappler, A. & Bryce, C. Cryoturbation impacts iron-organic carbon associations along a permafrost soil chronosequence in northern Alaska. *Geoderma* **413** (2022).
- Karhu, K. et al. Similar temperature sensitivity of soil mineral-associated organic carbon regardless of age. *Soil Biol. Biochem.* **136**, 107527 (2019).
- Kleber, M., Bourg, I. C., Coward, E. K., Hansel, C. M., Myneni, S. C. B. & Nunan, N. Dynamic interactions at the mineral-organic matter interface. *Nat. Rev. Earth*

- Environ.* **2**, 402-421 (2021).
- Lajtha, K., Townsend, K. L., Kramer, M. G., Swanston, C., Bowden, R. D. & Nadelhoffer, K. Changes to particulate versus mineral-associated soil carbon after 50 years of litter manipulation in forest and prairie experimental ecosystems. *Biogeochemistry* **119**, 341-360 (2014).
- Lavallee, J. M., Conant, R. T., Paul, E. A. & Cotrufo, M. F. Incorporation of shoot versus root-derived ¹³C and ¹⁵N into mineral-associated organic matter fractions: Results of a soil slurry incubation with dual-labelled plant material. *Biogeochemistry* **137**, 379-393 (2018).
- Lavallee, J. M., Soong, J. L. & Cotrufo, M. F. Conceptualizing soil organic matter into particulate and mineral-associated forms to address global change in the 21st century. *Glob. Chang. Biol.* **26**, 261-273 (2020).
- Liu, F. et al. Reduced quantity and quality of SOM along a thaw sequence on the Tibetan Plateau. *Environ. Res. Lett.* **13**, 104017 (2018).
- Lugato, E., Lavallee, J. M., Haddix, M. L., Panagos, P. & Cotrufo, M. F. Different climate sensitivity of particulate and mineral-associated soil organic matter. *Nat. Geosci.* **14**, 295-300 (2021).
- Ma, X., Asano, M., Tamura, K., Zhao, R. & Wang, T. Physicochemical properties and micromorphology of degraded alpine meadow soils in the eastern Qinghai-Tibet Plateau. *Catena* **194**, 104649 (2020).
- Marin-Spiotta, E., Silver, W. L., Swanston, C. W. & Ostertag, R. Soil organic matter dynamics during 80 years of reforestation of tropical pastures. *Glob. Chang. Biol.* **15**, 1584-1597 (2009).
- McCalley, C. K. et al. Methane dynamics regulated by microbial community response to permafrost thaw. *Nature* **514**, 478-481 (2014).
- McGuire, A. D. et al. Dependence of the evolution of carbon dynamics in the northern permafrost region on the trajectory of climate change. *Proc. Natl Acad. Sci. USA* **115**, 3882-3887 (2018).
- Monhonval, A. et al. Iron redistribution upon thermokarst processes in the Yedoma

- domain. *Front. Earth Sci.* **9**, 703339 (2021).
- Mu, C. et al. Carbon loss and chemical changes from permafrost collapse in the northern Tibetan Plateau. *J. Geophys. Res. Biogeosci.* **121**, 1781-1791 (2016).
- Mu, C., Zhang, F., Mu, M., Chen, X., Li, Z. & Zhang, T. Organic carbon stabilized by iron during slump deformation on the Qinghai-Tibetan Plateau. *Catena* **187**, 104282 (2020).
- Mu, C. et al. Soil organic carbon stabilization by iron in permafrost regions of the Qinghai-Tibet Plateau. *Geophys. Res. Lett.* **43**, 10286-10294 (2016).
- Patzner, M. S. et al. Seasonal fluctuations in iron cycling in thawing permafrost peatlands. *Environ. Sci. Technol.* <https://doi.org/10.1021/acs.est.1c06937> (2022).
- Patzner, M. S. et al. Iron mineral dissolution releases iron and associated organic carbon during permafrost thaw. *Nat Commun* **11**, 6329 (2020).
- Pizano, C., Barón, A. F., Schuur, E. A. G., Crummer, K. G. & Mack, M. C. Effects of thermo-erosional disturbance on surface soil carbon and nitrogen dynamics in upland arctic tundra. *Environm. Res. Lett.* **9**, 075006 (2014).
- Prater, I. et al. From fibrous plant residues to mineral-associated organic carbon—the fate of organic matter in Arctic permafrost soils. *Biogeosciences* **17**, 3367-3383 (2020).
- Qin, S., Kou, D., Mao, C., Chen, Y., Chen, L. & Yang, Y. Temperature sensitivity of permafrost carbon release mediated by mineral and microbial properties. *Sci. Adv.* **7**, eabe3596 (2021).
- Rowley, M. C., Grand, S. & Verrecchia, É. P. Calcium-mediated stabilisation of soil organic carbon. *Biogeochemistry* **137**, 27-49 (2018).
- Schuur, E. A. G. et al. Vulnerability of permafrost carbon to climate change: Implications for the global carbon cycle. *Bioscience* **58**, 701-714 (2008).
- Schuur, E. A. G. et al. Climate change and the permafrost carbon feedback. *Nature* **520**, 171-179 (2015).
- Sokol, N. W. & Bradford, M. A. Microbial formation of stable soil carbon is more

- efficient from belowground than aboveground input. *Nat. Geosci.* **12**, 46-53 (2018).
- Sokol, N. W., Sanderman, J. & Bradford, M. A. Pathways of mineral-associated soil organic matter formation: Integrating the role of plant carbon source, chemistry, and point of entry. *Glob. Chang. Biol.* **25**, 12-24 (2019).
- Stacy, E. M. et al. Stabilization mechanisms and decomposition potential of eroded soil organic matter pools in temperate forests of the Sierra Nevada, California. *J. Geophys. Res. Biogeosci.* **124**, 2-17 (2019).
- Todd-Brown, K. E. O. et al. Changes in soil organic carbon storage predicted by Earth system models during the 21st century. *Biogeosciences* **10**, 18969-19004 (2014).
- Vance, E. D., Brookes, P. C. & Jenkinson, D. S. An extraction method for measuring soil microbial biomass C. *Soil Biol. Biochem.* **19**, 703-707 (1987).
- von Lützw, M., Kögel-Knabner, I., Ekschmitt, K., Matzner, E. & Flessa, H. Stabilization of organic matter in temperate soils: mechanisms and their relevance under different soil conditions – a review. *Eur. J. Soil Sci.* **57**, 426-445 (2006).
- Vogel, C. et al. Clay mineral composition modifies decomposition and sequestration of organic carbon and nitrogen in fine soil fractions. *Biol. Fertility Soils* **51**, 427-442 (2015).
- Wiesmeier, M. et al. Aggregate stability and physical protection of soil organic carbon in semi-arid steppe soils. *Eur. J. Soil Sci.* **63**, 22-31 (2012).
- Wiesmeier, M. et al. Soil organic carbon storage as a key function of soils-A review of drivers and indicators at various scales. *Geoderma* **333**, 149-162 (2019).
- Wiltshire, S. E. Identification of homogeneous regions for flood frequency analysis. *J. Hydrol.* **84**, 287-302 (1986).
- Wu, J., Joergensen, R. G., Pommerening, B., Chaussod, R. & Brookes, P. C. Measurement of soil microbial biomass-C by fumigation-extraction-an automated procedure. *Soil Biol. Biochem.* **22**, 1167-1169 (1990).
- Yang, et al. Large-scale pattern of biomass partitioning across China's grasslands.

Global Ecol. Biogeogr. **19**, 268-277 (2010).

Yuan, M. M. et al. Microbial functional diversity covaries with permafrost thaw-induced environmental heterogeneity in tundra soil. *Glob. Chang. Biol.* **24**, 297-307 (2018).

Zou, D. et al. A new map of permafrost distribution on the Tibetan Plateau. *Cryosphere* **11**, 2527-2542 (2017).

SPRINGER NATURE
Author Services Editing Certificate

This document certifies that the manuscript

Divergent changes in particulate and mineral-associated organic carbon upon permafrost thaw

prepared by the authors

Futing Liu, Shuqi Qin, Kai Fang, Leiyi Chen, Yunfeng Peng, Pete Smith, Yuanhe Yang

was edited for proper English language, grammar, punctuation, spelling, and overall style by one or more of the highly qualified native English speaking editors at SNAS.

This certificate was issued on **June 14, 2022** and may be verified on the SNAS website using the verification code **3F00-E3E7-746D-0F73-18DB**.

Neither the research content nor the authors' intentions were altered in any way during the editing process. Documents receiving this certification should be English-ready for publication; however, the author has the ability to accept or reject our suggestions and changes. To verify the final SNAS edited version, please visit our verification page at secure.authorservices.springernature.com/certificate/verify.
If you have any questions or concerns about this edited document, please contact SNAS at support@as.springernature.com.

SNAS provides a range of editing, translation, and manuscript services for researchers and publishers around the world. For more information about our company, services, and partner discounts, please visit authorservices.springernature.com.

REVIEWERS' COMMENTS

Reviewer #2 (Remarks to the Author):

Dear Authors,

many thanks for the thorough revision of your manuscript and the extensive response to my remarks and concerns. I'm looking forward to see the work published.

With kind regards

Responses to Reviewer #2

[Comment] Dear Authors, many thanks for the thorough revision of your manuscript and the extensive response to my remarks and concerns. I'm looking forward to see the work published. With kind regards.

[Response] Thanks for the reviewer's positive comments and recognition for our revision.